# Calcium-mediated actin reset (CaAR) mediates acute cell adaptations

Pauline Wales[1,2†], Christian E Schuberth[1,2†], Roland Aufschnaiter[1,2†], Johannes Fels[1,2], Ireth García-Aguilar[3], Annette Janning[1,2], Christopher P Dlugos[1,2,4], Marco Schäfer-Herte[1,2], Christoph Klingner[1,2,5], Mike Wälte[1,2], Julian Kuhlmann[1,2], Ekaterina Menis[1,2], Laura Hockaday Kang[1,2], Kerstin C Maier[6], Wenya Hou[7], Antonella Russo[8], Henry N Higgs[9], Hermann Pavenstädt[4], Thomas Vogl[8], Johannes Roth[8], Britta Qualmann[7], Michael M Kessels[7], Dietmar E Martin[6], Bela Mulder[3], Roland Wedlich-Söldner[1,2*]

[1]Institute of Cell Dynamics and Imaging, University of Muenster, Muenster, Germany; [2]Cells-In-Motion Cluster of Excellence (EXC1003 – CiM), University of Münster, Muenster, Germany; [3]Theory of Biological Matter, FOM Institute AMOLF, Amsterdam, Netherlands; [4]Medical Clinic D, University Clinic of Muenster, Muenster, Germany; [5]AG Molecular Mechanotransduction, Max Planck Institute of Biochemistry, Munich, Germany; [6]Department of Biochemistry, University of Munich, Munich, Germany; [7]Institute of Biochemistry I, Friedrich Schiller University Jena, Jena, Germany; [8]Institute of Immunology, University of Münster, Münster, Germany; [9]Department of Biochemistry, Dartmouth Medical School, Hanover, United States

*For correspondence: wedlich@ uni-muenster.de

†These authors contributed equally to this work

Competing interests: The authors declare that no competing interests exist.

**Abstract** Actin has well established functions in cellular morphogenesis. However, it is not well understood how the various actin assemblies in a cell are kept in a dynamic equilibrium, in particular when cells have to respond to acute signals. Here, we characterize a rapid and transient actin reset in response to increased intracellular calcium levels. Within seconds of calcium influx, the formin INF2 stimulates filament polymerization at the endoplasmic reticulum (ER), while cortical actin is disassembled. The reaction is then reversed within a few minutes. This Calcium-mediated actin reset (CaAR) occurs in a wide range of mammalian cell types and in response to many physiological cues. CaAR leads to transient immobilization of organelles, drives reorganization of actin during cell cortex repair, cell spreading and wound healing, and induces long-lasting changes in gene expression. Our findings suggest that CaAR acts as fundamental facilitator of cellular adaptations in response to acute signals and stress.

## Introduction

Actin organization and dynamics are critical for most morphogenetic processes, including cell polarization, migration and division. The prominent role of the actin cytoskeleton is reflected in a host of regulators that mediate dynamic assembly of complex actin structures from filament bundles and networks. These structures in turn provide protrusive and contractile forces during physiological and pathological processes such as differentiation, wound healing and tumor metastasis (*Lecuit et al., 2011*; *Pantaloni et al., 2001*; *Pollard and Cooper, 2009*). While many studies in the past have focused on the molecular characterization of specific actin assemblies and their local function within cells, recent work has elegantly shown that a balance between different actin assemblies is established in cells by distinct actin nucleators. These nucleators constantly compete for a common pool

**eLife digest** Our skeleton plays a vital role in giving shape and structure to our body, it also allows us to make coordinated movements. Similarly, each cell contains a microscopic network of structures and supports called the cytoskeleton that helps cells to adopt specific shapes and is crucial for them to move around. Unlike our skeleton, which is relatively unchanging, the cytoskeleton of each cell constantly changes and adapts to the specific needs of the cell.

One part of the cytoskeleton is a dense, flexible meshwork of fibers called the cortex that lies just beneath the surface of the cell. The cortex is constructed using a protein called actin, and many of these proteins join together to form each fiber. When cells need to adapt rapidly to an injury or other sudden changes in their environment they activate a so-called stress response. This response often begins with a rapid increase in the amount of calcium ions inside a cell, which can then trigger changes in actin organization. However, it is not clear how cells under stress are able to globally remodel their actin cytoskeleton without compromising stability and integrity of the cortex.

Wales, Schuberth, Aufschnaiter et al. used a range of mammalian cells to investigate how actin responds to stress signals. All cells responded to the resulting influx of calcium ions by deconstructing large parts of the actin cortex and simultaneously forming actin filaments near the center of the cell. Wales, Schuberth, Aufschnaiter et al. termed this response calcium-mediated actin reset (CaAR), as it lasted for only a few minutes before the actin cortex reformed. The experiments show that a protein called INF2 controls CaAR by rapidly removing actin from the cortex and forming new filaments near a cell compartment called the endoplasmic reticulum.

CaAR allows cells to rapidly and drastically alter the cortex in response to stress. The experiments also show that this sudden shift in actin can change the activity of certain genes, leading to longer-term effects on the cell. The findings of Wales, Schuberth, Aufschnaiter et al. suggest that calcium ions globally regulate the actin cytoskeleton and hence cell shape and movement under stress. This could be relevant for many important processes and conditions such as wound healing, inflammation and cancer. A future challenge will be to understand the role of CaAR in these processes.

of actin monomers (*Burke et al., 2014*), and the balance between actin assemblies can be tightly controlled by regulatory factors such as profilin (*Rotty et al., 2015*; *Suarez et al., 2015*). In addition, cellular F- and G-actin levels have been linked to profound and long-lasting changes in cell physiology through regulation of transcriptional cofactors such as myocardin-related transcription factors (MRTF), the Yes-associated protein (YAP) or transcriptional co-activator with PDZ-binding motif (TAZ) (*Halder et al., 2012*; *Miralles et al., 2003*). Finally, the rapidly expanding field of mechanobiology has highlighted global integration of actomyosin-mediated forces across the cortex of single cells (*Klingner et al., 2014*; *Salbreux et al., 2012*), as well as within cell groups and whole tissues (*Lecuit et al., 2011*; *Trepat et al., 2012*).

In the context of widespread interdependence, competition and mechanical connectivity between actin assemblies, cells face formidable challenges when undergoing a global reorganization of their actin cytoskeleton. This is exemplified by the impact of moderate perturbations of the cortical actin cytoskeleton with low doses of latrunculin, which can result in a massive imbalance of actomyosin-mediated forces and lead to large-scale fluctuations in cell cortex organization (*Klingner et al., 2014*; *Luo et al., 2013*). How cells globally coordinate actin remodeling during rapid and profound morphological transitions has not been sufficiently addressed. Shear flow has been shown to induce rapid changes in cellular actin organization and mechanics (*Rahimzadeh et al., 2011*; *Verma et al., 2012*). In addition, a recent study reported calcium-mediated rapid and transient formation of actin filaments at the nuclear periphery of fibroblasts exposed to mechanical stress (*Shao et al., 2015*). However, how these observations can be linked to cell cortex organization and overall cell physiology is not known.

Here we show that calcium not only induces actin polymerization at the nuclear periphery, but causes a rapid reduction of cortical actin coinciding with the formation of new filaments at the entire endoplasmic reticulum (ER). This global change in actin distribution is rapidly reversed within less than two minutes. Calcium-mediated Actin Reset (CaAR) occurs in a wide range of mammalian cell

types and is initiated by a variety of physiological signals. Most importantly, we found that CaAR acts as a fundamental facilitator of acute cell responses. It is required for repair of cortical damage and coordinates the formation of actin-rich protrusions during cell spreading and cell migration. In addition, CaAR induces long-term changes in gene expression via MRTF and serum response factor (SRF). Our results have important implications for a multitude of experiments where cells are exposed to acute stress as well as for calcium-regulated processes such as wound healing, inflammation, cell differentiation and cancer progression.

## Results

### A calcium-mediated actin reset in mammalian cells

To establish the cellular response to an acute mechanical stimulus we exposed Madin-Darby Canine Kidney (MDCK) epithelial cells to sudden shear flow. Prior to shear stress, MDCK cells stably expressing Lifeact-GFP exhibited a typical apical actin organization with clustered microvilli (*Klingner et al., 2014*). Immediately upon exposure to fluid flow of 10–20 dyn/cm$^2$ we observed the formation of a highly transient perinuclear actin ring, which only remained for a few minutes (*Figure 1A*, *Video 1*). A similar phenomenon was recently reported in fibroblasts (*Shao et al., 2015*), but its consequences for cell organization were not further explored. To establish the reported relevance of calcium (*Shao et al., 2015*) for the actin reorganization in MDCK cells, we monitored Ca$^{2+}$ levels with the fluorescent probe Fluo-4. Upon induction of shear stress, actin reorganization was preceded by a strong intracellular Ca$^{2+}$ pulse (*Figure 1A,B*). To examine whether the elevation of Ca$^{2+}$ levels is causally linked to actin reorganization, we treated MDCK cells with the calcium ionophore ionomycin. Indeed, exposure to as little as 0.3 µM ionomycin induced the formation of perinuclear actin rings and reduction of cortical actin within 60 s (*Figure 1C*). Interestingly, in contrast to the previous report, we found a virtually simultaneous decrease of actin filaments at the apical cortex (*Figure 1C*). As for the transient shear stress response, actin rapidly reverted to its cortical distribution within the following 60 s (*Figure 1C*). To test the prevalence of calcium-mediated actin reorganization, we stably or transiently expressed Lifeact-GFP in a panel of mammalian cell lines, including epithelial, mesenchymal, endothelial and immune cells, and treated each with ionomycin. In all instances, rapid relocation of actin from the cell cortex to the nuclear periphery occurred within 60 s (*Figure 1—figure supplement 1A*, *Video 2*). We observed the strongest response for human MCF-7 breast cancer cells (*Figure 1D*, *Video 3*) and therefore decided to focus on this cell line for further studies. Actin reorganization in MCF-7 cells was not influenced by Lifeact-GFP expression, as we observed an identical response in non-transfected cells stained with Alexa Fluor 647-phalloidin (*Figure 1—figure supplement 1B*). Importantly, we found that, upon calcium influx, actin filaments formed not only at the nuclear periphery, but throughout the cell (*Figure 1—figure supplement 1C*) and all along the endoplasmic reticulum (ER, *Figure 1E*). If cytosolic calcium is the key regulator of rapid actin reorganization, we reasoned that other signals that induced sufficient Ca$^{2+}$ influx should be able to elicit the actin response. As expected, activation of calcium influx using physiological ligands for G-protein-coupled receptors, such as ATP or bradykinin efficiently induced actin rearrangement (*Figure 1F*). In most cells release of calcium from ER stores activates store-operated calcium entry from the extracellular environment (*Hogan and Rao, 2015*). Hence, blocking uptake of Ca$^{2+}$ into the ER with thapsigargin - in the presence of high extracellular Ca$^{2+}$ - also induced actin rearrangement (*Figure 1F*). Finally, perforation of the plasma membrane by localized mechanical disruption (atomic force microscopy, AFM) or laser-induced ablation efficiently induced cortex to ER actin reorganization (*Figure 1F*).

In summary, we have identified a fundamental and conserved process of rapid, global and transient actin rearrangement, which occurs in a wide range of mammalian cells and can be induced by a variety of signals that raise cytosolic calcium levels. Due to the striking inversion of cellular actin organization that we observed (cortex-ER-cortex, *Figure 1C,D*) we decided to term this process 'Calcium-mediated Actin Reset' or CaAR.

### Quantitative analysis of CaAR

We proceeded to characterize the detailed kinetics and features of CaAR to gain insights into the underlying molecular mechanisms. In all cell types that we tested, CaAR was initiated within seconds

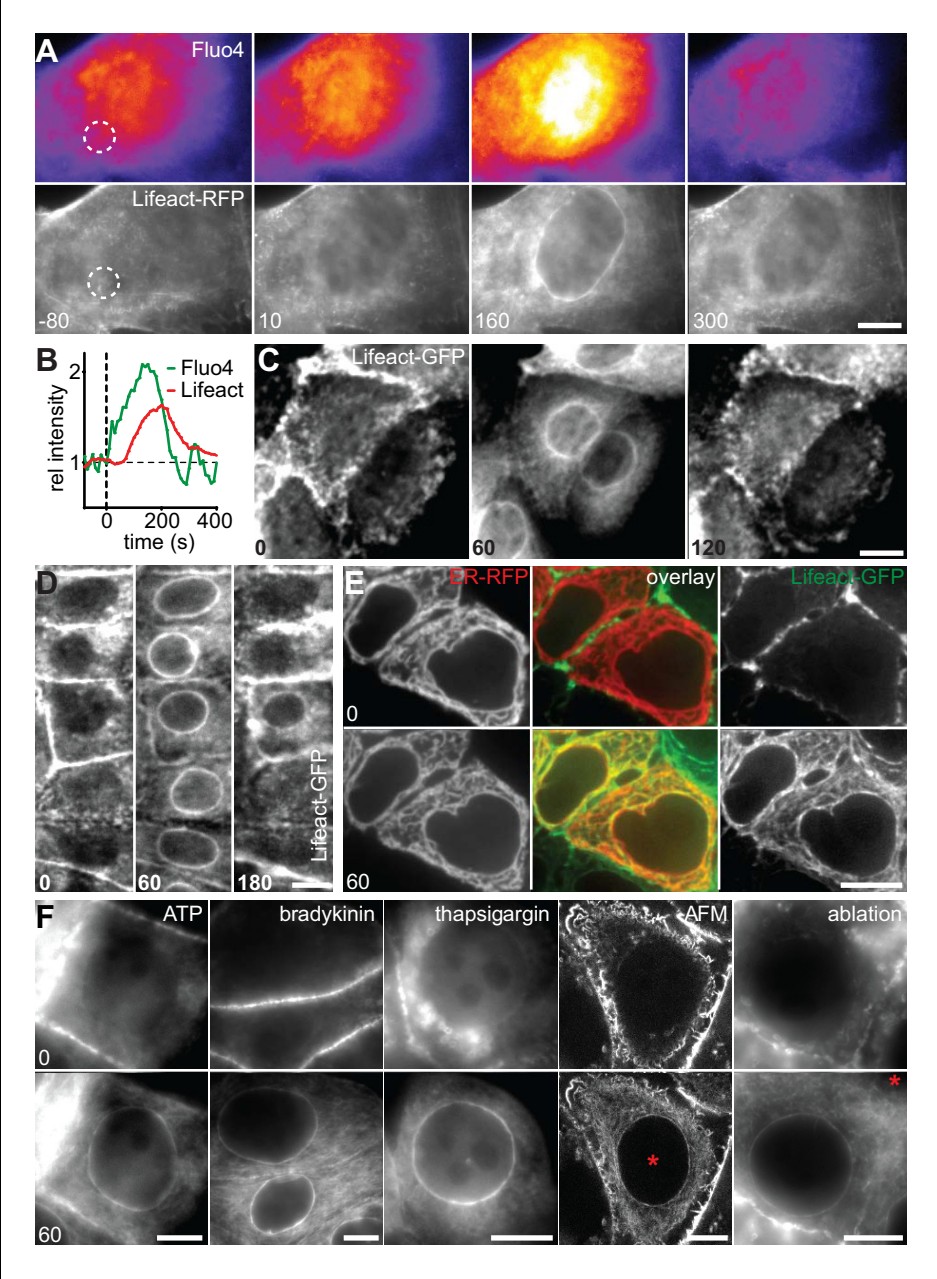

**Figure 1.** A calcium-mediated actin reset in mammalian cells. (**A, B**) MDCK cells labeled with Lifeact-mCherry (Lifeact-RFP) and Fluo4 were exposed to 10 dyn/cm$^2$ shear flow. Regions used for intensity plots in (**B**) are indicated in (**A**). (**C**) MDCK cells expressing Lifeact-GFP were stimulated with 1 µM ionomycin. (**D–F**) MCF-7 cells expressing Lifeact-GFP (and ER-RFP in (**E**)) were exposed to 1 µM ionomycin (**D, E**), or to 50 µM ATP, 1 µM bradykinin or 1 µM thapsigargin or locally stressed with an AFM probe (pointy tips) or by laser ablation (**F**) (asterisks at position of stimulus). Times in sec. Scale bars: 10 µm.

The following figure supplement is available for figure 1:

**Figure supplement 1.** A calcium-mediated actin reset in mammalian cells.

---

after Ca$^{2+}$ influx (*Figure 2—figure supplement 1A* and not shown). Actin concentration at the ER reached its maximum within less than 1 min and reverted back to the original state within less than 5 min. Actin recruitment to the ER followed a typical three-phase kinetics characterized by a rapid

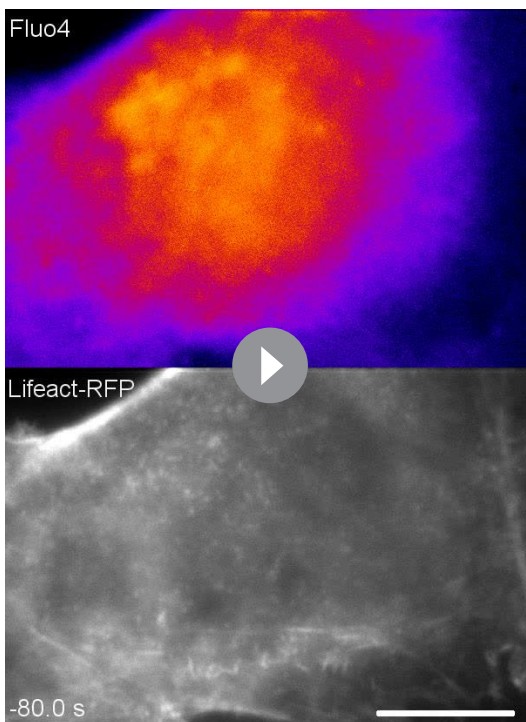

**Video 1.** MDCK cells expressing Lifeact-mCherry and labeled with Fluo4 exposed to shear flow (10 dyn/cm²). Corresponds to *Figure 1A*. Scale bar: 10 µm.

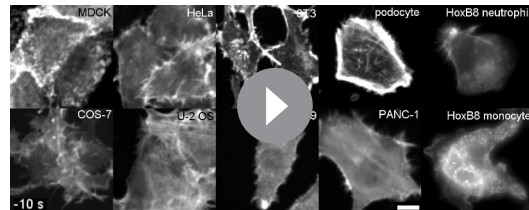

**Video 2.** A panel of indicated cell types expressing Lifeact-GFP exposed to 1 µM ionomycin. Corresponds to *Figure 1—figure supplement 1A*. Scale bar: 10 µm.

increase, a plateau and a slower decrease (*Figure 2A*). We considered serum-starved (for 1 hr) MCF-7 cells treated with 330 nM ionomycin at RT as a control condition. The actin increase at the ER was significantly faster for cells grown in serum or at 37°C. Also, the stimulation of MCF-7 and HeLa cells with 50 µM ATP led to faster increase than for ionomycin (*Figure 2B*, *Table 1*). In contrast, addition of serum or ATP had little effect on the rate of decrease or the amplitude of the reaction (*Figure 2B*). Finally, the CaAR plateau was shortened at 37°C, for cells grown in serum and upon stimulation with ATP (*Figure 2B*). Treatment of HeLa cells with ionomycin induced very strong and long-lasting CaAR response (*Figure 2B*), indicating that these cells could not as efficiently remove Ca²⁺ from the cytosol. Most importantly, despite the specific differences described above, all kinetic parameters of CaAR remained within a three-fold range (*Figure 2B*), highlighting the robust and stereotypic nature of the response.

Next, we investigated the role of calcium in more detail using the ratiometric dye Fura2. We found that cytosolic Ca²⁺ levels in MCF-7 cells increased 2- to 4-fold upon exposure to ionomycin (3.59 ± 0.84 fold, n = 31) or ATP (2.82 ± 0.67 fold, n = 54), which induced CaAR in virtually all cells (*Figure 2C*). Induction of CaAR was completely prevented in Ca²⁺-free medium, but could be restored within 1 min by addition of Ca²⁺ (not shown). In Ca²⁺-free medium, release of calcium from ER stores with thapsigargin could not be enhanced by store-operated calcium entry and led to a modest cytosolic Ca²⁺ increase of 1.26 ± 0.08 fold (n = 31). This was insufficient to induce CaAR (*Figure 2C*). These results indicate that CaAR induction requires a threshold level of intracellular Ca²⁺ that can only be reached by influx from the extracellular environment.

Observing such large-scale reorganization of actin, we wondered whether the mechanical properties of cells undergoing CaAR were altered. We therefore used atomic force microscopy (AFM) to probe cells with 10 µm beads attached to the cantilever. Despite the observed transient reduction of cortical actin, we found that the subcortical regions (800 nm below the plasma membrane) of MCF-7 cells treated with ionomycin became markedly stiffer during CaAR, mirroring greater levels of actin recruitment at the ER (*Figure 2D*). In addition, we observed that intracellular motility of organelles was transiently

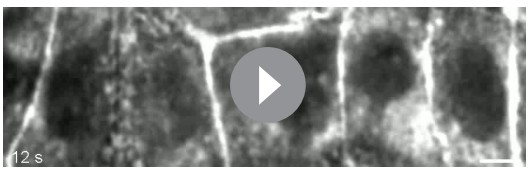

**Video 3.** MCF-7 cells expressing Lifeact-GFP exposed to 1 µM ionomycin. Corresponds to *Figure 1D*. Scale bar: 10 µm.

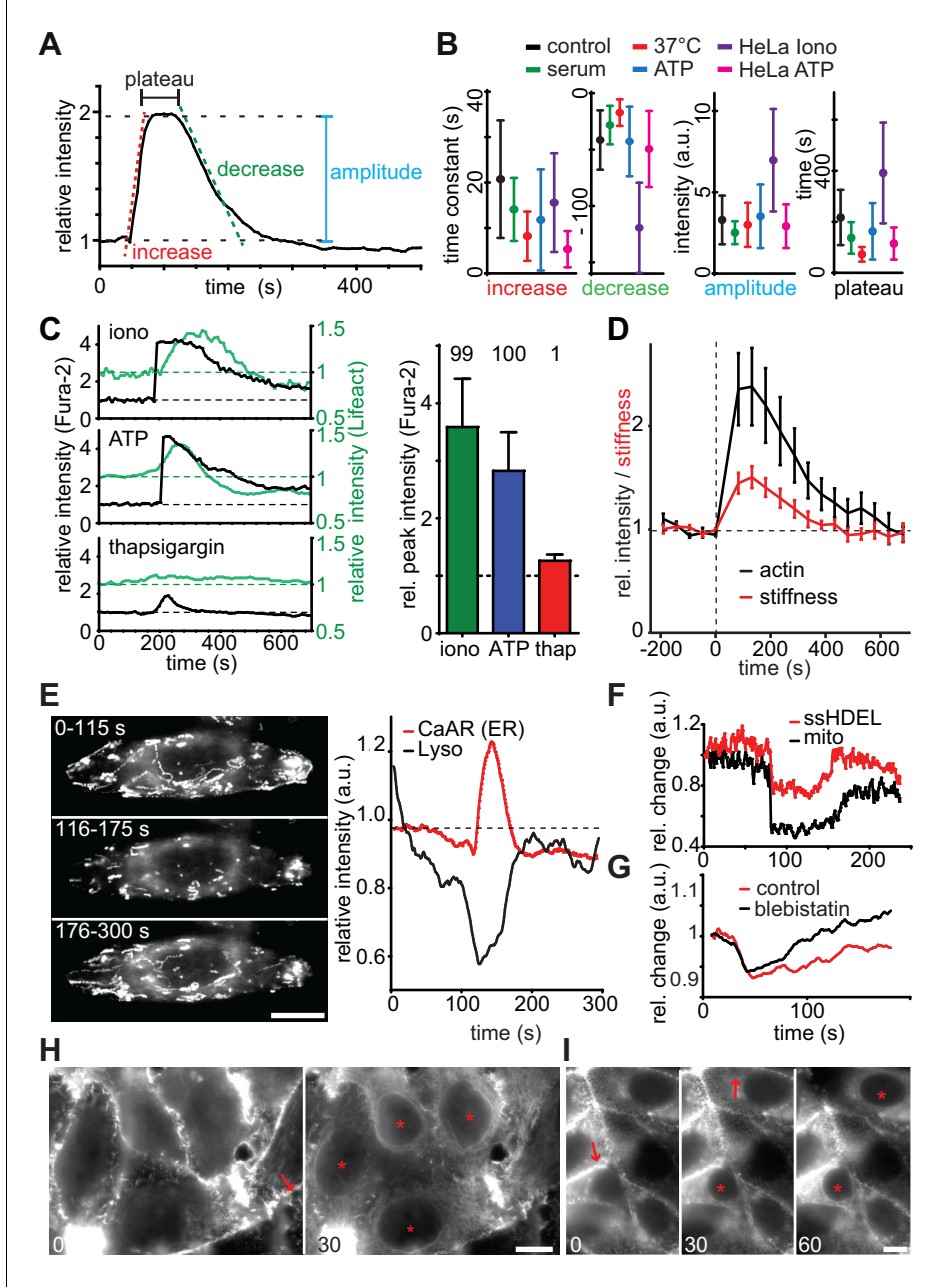

**Figure 2.** Quantitative analysis of CaAR. (**A, B**) Quantification of indicated parameters during CaAR. MCF-7 and HeLa cells expressing Lifeact-GFP were incubated at RT or 37°C and with serum or serum-free HBSS buffer. Cells were then treated with either 330 nM ionomycin or 50 µM ATP. Control corresponds to serum starved MCF-7 cells stimulated with 330 nM ionomycin at RT. GFP intensity was measured at the nuclear periphery and analyzed using customized Matlab scripts (Supplementary material). All values are mean ± SD, n > 100 cells. See *Table 1* for details. (**C**) Lifeact-GFP expressing MCF-7 cells were treated with the indicated drugs, ATP: 50 µM, ionomycin: 1 µM, thapsigargin: 1 µM in $Ca^{2+}$-free medium. Intracellular calcium levels (Fura2) and Lifeact-GFP intensities at the ER were monitored. Peak values: mean ± SD (n > 30). Numbers above bars indicate % cells exhibiting CaAR. (**D**) Cells treated with 1 µM ionomycin were followed over time by simultaneous fluorescence and atomic force microscopy. The relative Young's modulus of whole cells was calculated from force-distance curves obtained with 10 µm beads (mean ± SD, n = 24). (**E**) Freezing of lysosomes labeled with Lysotracker Red in HeLa cells undergoing CaAR. Images correspond to the maximum projection of indicated periods in a time series. (**F**) Freezing of organelle motion during CaAR in MCF-7 cells. ER or mitochondria were fluorescently labeled with ss-RFP-KDEL or mitotracker Red, respectively. (**G**) Change in lysosome motility for control and blebbistatin-treated (50 µM) HeLa cells. (**H, I**) Propagation of CaAR induced by laser ablation in the absence (**I**) and presence (**J**) of 50

*Figure 2 continued on next page*

*Figure 2 continued*

µM ATP. Arrows: ablation sites, asterisks: cells reacting to stimulus. Times in sec after exposure to the stimulus. Scale bars: 10 µm.

The following figure supplement is available for figure 2:

**Figure supplement 1.** Quantitative analysis of CaAR.

halted during CaAR. Random as well as directed motion of lysosomes (*Figure 2E*), mitochondria and the ER network (*Figure 2F*) was abolished within a few seconds of CaAR onset and resumed after actin returned to the cell cortex (*Figure 2E,F*). When simultaneously observing CaAR and lysosome motility we found the organelles trapped within a cytosol-filling actin mesh that corresponded to the transient ER-based actin filaments (*Video 4*). Lysosomes were also immobilized in cells treated with 50 µM blebbistatin arguing against a prominent role of actomyosin contractility for organelle freezing (*Figure 2G*).

When using pointy AFM tips to probe cells, we often were able to induce CaAR in individual cells (*Figure 1E*, *Figure 2—figure supplement 1B,C*). Forces above 50 nN (*Figure 2—figure supplement 1B*) were able to robustly induce CaAR multiple times in a single cell (*Figure 2—figure supplement 1D*, *Video 5*) with a refractory period of 30–60 s (*Figure 2—figure supplement 1C*), consistent with the time scale of actin recruitment at the ER. When using localized CaAR stimulation by AFM we frequently observed that cells directly adjacent to the manipulated cells also reacted (*Figure 2—figure supplement 1E*). To examine this behavior in more detail, we performed ablation experiments on MCF-7 monolayers. Strikingly, we found that nearly all cells within 60–150 µm of the ablation site exhibited CaAR (*Figure 2H*). This was also observed for cells that were not in direct contact with the ablated cell (*Figure 2—figure supplement 1F*, *Video 6*). Previous reports have shown that $Ca^{2+}$ signals can be propagated across tissue sections and cell layers via ATP

**Table 1.** Quantitative analysis of CaAR.

| Cell type | MCF-7 | MCF-7 | MCF-7 | MCF-7 | HeLa | HeLa | |
|---|---|---|---|---|---|---|---|
| condition | control | serum | 37°C | ATP | Iono | ATP | |
| | 20,74 | 14,09 | 8,20 | 11,78 | 15,59 | 5,34 | mean |
| increase time constant (s) | 12,94 | 6,94 | 5,42 | 11,12 | 10,86 | 3,99 | stdev |
| | 980 | 588 | 102 | 252 | 371 | 258 | n |
| | | **** | **** | **** | **** | **** | **** | ANOVA |
| | −41,47 | −28,21 | −17,15 | −42,74 | −119,30 | −49,42 | mean |
| decrease time constant (s) | 26,32 | 17,06 | 11,75 | 30,90 | 40,00 | 33,71 | stdev |
| | 1010 | 586 | 102 | 279 | 136 | 193 | n |
| | | **** | **** | ns | **** | ns | ANOVA |
| | 3,29 | 2,51 | 2,99 | 3,52 | 6,97 | 2,91 | mean |
| amplitude intensity (a.u.) | 1,50 | 0,71 | 1,36 | 1,96 | 3,15 | 1,34 | stdev |
| | 979 | 588 | 102 | 252 | 371 | 255 | n |
| | | **** | ns | ns | **** | *** | ANOVA |
| | 217,30 | 137,00 | 72,65 | 162,70 | 391,40 | 115,20 | mean |
| plateau time (s) | 108,20 | 61,49 | 28,60 | 110,50 | 197,40 | 62,77 | stdev |
| | 965 | 586 | 102 | 238 | 136 | 193 | n |
| | | **** | **** | **** | **** | **** | ANOVA |

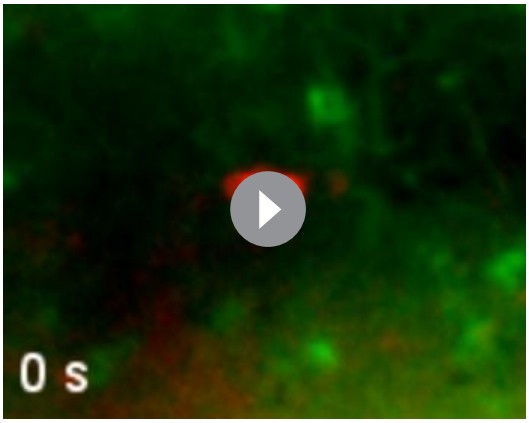

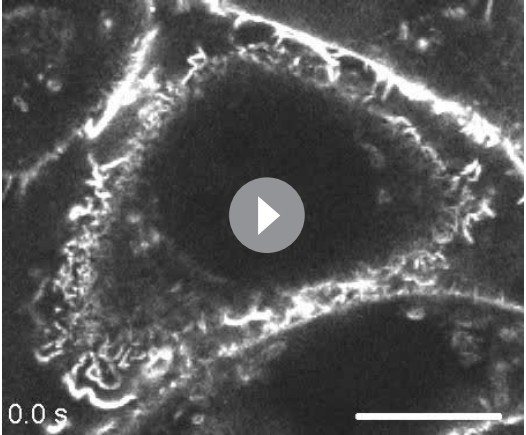

**Video 4.** HeLa cell expressing Lifeact-GFP labelled with Lysotracker Red and stimulated by laser ablation. Corresponds to *Figure 2E*.

**Video 5.** MCF-7 cell expressing Lifeact-GFP repeatedly stimulated by AFM (asterisk). Corresponds to *Figure 2—figure supplement 1D*. Scale bar: 10 µm.

(*Frame and de Feijter, 1997*; *Schwiebert, 2000*). Indeed, distant cells were no longer able to respond to cell ablation after the medium had been saturated with 100 µM ATP (*Figure 2I*), indicating that ATP release and associated $Ca^{2+}$ influx is responsible for the propagation of CaAR.

## CaAR is driven by INF2-mediated actin polymerization

The rapid increase in actin localization at the ER indicated the involvement of a strong nucleator in CaAR. Accordingly, both G-actin sequestration by latrunculin A (LatA) and actin filament disruption by cytochalasin D (CytoD) completely blocked CaAR (*Figure 3A*, *Video 7*). The only actin nucleator that has been shown to be localized at the ER is inverted formin 2 (INF2) (*Chhabra and Higgs, 2006*), and this nucleator has also been shown to mediate the formation of perinuclear actin filaments in 3T3 fibroblasts (*Shao et al., 2015*). Indeed, in MCF-7 and HeLa cells, INF2 was localized to the ER, as shown either by immunofluorescence (*Figure 3B*, *Figure 3—figure supplement 1A*) or when expressing the full-length INF2-CAAX isoform fused to GFP (*Figure 3—figure supplement 1B*). A constitutively active point mutant of INF2 (A149D, (*Korobova et al., 2013*) induced the previously shown strong actin staining at the ER (*Ramabhadran et al., 2013*), which was not further increased by the addition of ionomycin (*Figure 3C*). To test a potential role of INF2 for CaAR we knocked down INF2 expression by RNA interference. While our attempts for INF2 knock down were unsuccessful in MCF-7 cells, we were able to efficiently abolish expression of INF2 in HeLa cells (*Figure 3D*). Strikingly, CaAR was completely blocked in HeLa cells that were depleted for INF2 (*Figure 3E*). This was especially apparent in areas where some cells still retained INF2 expression and therefore were still able to nucleate actin at the ER upon treatment with ionomycin (*Figure 3E*, circled area). Notably, INF2 knock-down also blocked actin decrease at the cell cortex (*Figure 3E*), increase in cell stiffness detected by AFM (*Figure 3—figure supplement 1C*) and freezing of organelle motility (*Figure 3—figure supplement 1D*. To obtain stable cell lines without INF2 expression we knocked out INF2 using the CRSPR/Cas9 system. We obtained several independent HeLa INF2-KO clones that exhibited no detectable INF2 expression by Western blot (*Figure 3—figure supplement 1E*). We then transiently expressed either the CAAX- or nonCAAX human isoform of INF2 in clone 18 (*Figure 3F*). Upon induction of CaAR by laser ablation we observe no reaction in untransfected cells confirming that INF2 is essential for CaAR. Interestingly, both isoforms were able to rescue the knock out and to support efficient CaAR reactions (*Figure 3F*), indicating that ER-localization was not essential for actin reorganization.

Considering the rapid activation of INF2 by calcium and the lack of obvious calcium binding sites in the formin itself, we hypothesized that an abundant calcium sensor protein could be involved in the reaction. We therefore investigated binding of INF2 to the prototypical calcium regulator calmodulin. Co-precipitation analyses showed that GFP-INF2 expressed in HEK293 cells indeed

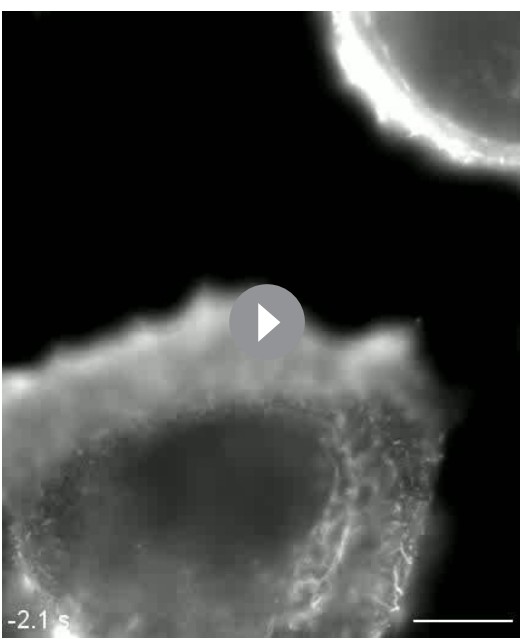

**Video 6.** Propagation of CaAR in MCF-7 cells expressing Lifeact-GFP stimulated by laser ablation (asterisk). Corresponds to *Figure 2—figure supplement 1F*. Scale bar: 10 µm.

specifically bound to immobilized calmodulin, and that this interaction was Ca²⁺-dependent (*Figure 3G*). While this result provides a potential molecular basis for INF2 activation, additional work will be needed to clarify the specific role of calmodulin during CaAR.

It is well documented that intracellular calcium influx also affects other cellular factors such as Myosin II, and INF2 has been linked to microtubule stabilization (*Andrés-Delgado et al., 2012*; *Bartolini et al., 2016*). These factors might therefore also play a role during CaAR. However, pretreatment of cells with the myosin II inhibitor blebbistatin or with the microtubule disrupting drug nocodazole did not inhibit or slow down CaAR (*Figure 3—figure supplement 1F*). In summary, our results indicate that CaAR is largely dependent on a single factor – the calcium-regulated actin nucleator INF2.

## A stochastic model rationalizes CaAR features and kinetics

Two remarkable characteristics of CaAR are the reciprocal behavior of actin at the ER and the cell cortex and the highly transient nature of the reorganization (*Figure 4A*). Interestingly, INF2 is not only an actin nucleator and elongator, but also a very potent severing and actin-depolymerizing factor (*Chhabra and Higgs, 2006*; *Gurel et al., 2014*). During CaAR, these activities should also be subject to regulation by calcium and therefore potentially facilitate the observed reciprocal and transient reaction. It is important to note that CaAR does not lead to the equal reduction of all cortical actin structures. Stable actin assemblies such as stress fibers were much less affected (*Figure 4—figure supplement 1A*, *Video 8*). Also, despite their opposite slopes, there was no apparent delay between ER and cell cortex reactions and they followed the calcium signal with the same offset (*Figure 4A*). In addition, the duration of Ca²⁺ influx was directly correlated with the kinetics of actin reorganization at both locations (*Figure 4B*, *Figure 4—figure supplement 1B*). These observations indicate that the actin filament assembly and disassembly at the ER and cell cortex are tightly coupled. A potential mechanism for such coupling would be simple competition between different actin nucleators. To test whether such a scenario could account for the observed CaAR kinetics, we developed a particle-based stochastic model of actin dynamics that explicitly considers actin nucleation, polymerization, depolymerization, capping and severing (*Figure 4C*, Materials and Methods). We initially focused on a single actin population in equilibrium with the monomer pool. The model showed that actin turnover is mainly determined by the speed of actin depolymerization and modulated to a lesser extent by the rates of capping and severing (*Figure 4—figure supplement 1C*). In addition, actin turnover was strongly dependent on the levels of actin monomers available (*Figure 4—figure supplement 1D*). The experimentally measured $t_{1/2}$ for recovery of actin-GFP at the cortex of MCF-7 cells was quite fast, at 14.48 ± 7.21 s (n = 79). This constrained the possible values for severing, depolymerization and ratios of F- to G-actin (*Figure 4—figure supplement 1D,E*). We next included a second nucleation activity to represent INF2-mediated actin polymerization at the ER (*Figure 4C*). By assuming that INF2 activity follows the intracellular Ca²⁺ curve (See Materials and Methods), we were able to generate a transient actin peak at the ER with a corresponding decrease at the cortex (*Figure 4D*). By increasing the INF2 nucleation rate or the amount of available G-actin in the system we were able to obtain kinetics that were reasonably close to the experimental observations (dotted curves in *Figure 4D*). However, cortical actin never dropped below 50% of its original value, and we were unable to obtain complete reorganization of actin within less than 30 s, as seen with laser ablation (*Figure 4B*). Interestingly, the model predicted that including Ca²⁺- or INF2-dependent actin

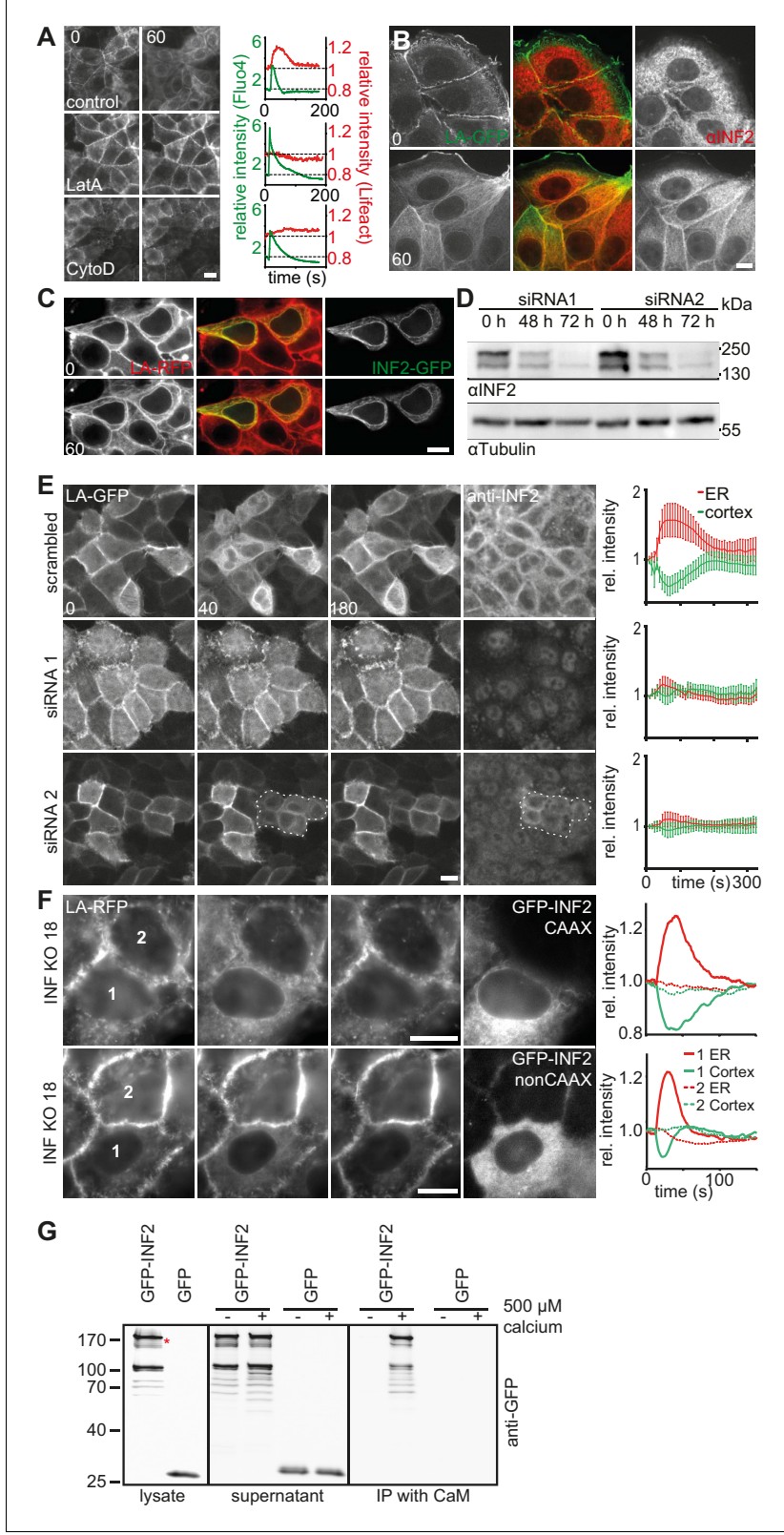

**Figure 3.** CaAR is driven by INF2-mediated actin polymerization. (**A**) MCF-7 cells expressing Lifeact-mCherry were treated with ionomycin and the indicated drugs (LatA: 400 nM latrunculin A, CytoD: 1 μM cytochalasin D). Plots show representative intensity profiles for Lifeact-GFP at the nuclear periphery and for intracellular Ca²⁺ (Fluo4). (**B**) MCF-7 cells were fixed at the indicated time points after addition of 1 μM ionomycin and stained with αINF2
*Figure 3 continued on next page*

*Figure 3 continued*

antibody. (**C**) MCF-7 cells expressing Lifeact-mCherry were transfected with a constitutively active GFP-INF2 (A149D) construct and imaged before and after addition of 1 µM ionomycin. (**D, E**) siRNA-mediated knock-down of INF2 in HeLa cells expressing Lifeact-GFP. (**D**) Western analysis of INF2 after knock-down with two different siRNAs at indicated times. (**E**) 72 hr after siRNA transfection, HeLa cells were treated with 1 µM ionomycin and monitored by fluorescence microscopy. Cells were fixed, immunostained with anti-INF2 antibody and imaged again at the same positions. Dotted line surrounds residual INF2-positive cells. Plots show Lifeact-GFP intensity at the cortex (green) and the ER (red). Values are mean ± SD, n = 30. (**F**) Images of HeLa INF2 KO cells stably expressing Lifeact-mCherry (derived from KO clone 18) and transfected with either GFP-INF2-CAAX or GFP-INF2-nonCAAX. Cells were stimulated by laser ablation outside the represented region. In each series one cell with INF2 expression (1) and a control cell without INF2 (2) are labeled. Corresponding intensity plots for ER (red) and cortical (green) regions are shown. Times in sec. Scale bars: 10 µm. (**G**) Co-precipitation analyses showing that GFP-INF2-CAAX expressed in HEK293 cells specifically interacts with immobilised calmodulin (CaM). Comparison of pull down conditions with and without (1 mM EGTA) 500 µM $Ca^{2+}$ (CaM activation at plateau).

The following figure supplement is available for figure 3:

**Figure supplement 1.** CaAR is driven by INF2-mediated actin polymerization.

---

depolymerization into the model strongly increased the amplitude and speed of actin reorganization at both cortex and ER (solid curves in *Figure 4D*). To experimentally test this prediction we measured CaAR kinetics in HeLa KO cells expressing a WH2-mutant of INF2 (3L: L976A, L977A, L986A) that was reported to have strongly reduced depolymerization activity (*Chhabra and Higgs, 2006*). Strikingly CaAR kinetics were strongly reduced in cells expressing the 3L mutant. The resulting changes were comparable to the situation without $Ca^{2+}$-mediated depolymerization predicted by the model (*Figure 4D,E*). The amplitude and the speed of reorganization were strongly reduced for both, ER and cell cortex (*Figure 4F,G*).

In summary, our simulations indicate that the observed kinetics of CaAR can be quantitatively explained by a competition scenario with constitutively active cortical actin nucleators and $Ca^{2+}$-activated INF2 at the ER. Importantly, the depolymerization activity of INF2 provides additional optimization for CaAR kinetics.

## CaAR induces transcription changes

While the most apparent feature of CaAR is certainly the transient and reciprocal reorganization of actin, we wondered whether the global balance between G- and F-actin was also affected during or after the reaction. Indeed, fractionation of MCF-7 cell extracts revealed a modest increase in the global F/G-actin ratio during CaAR (*Figure 5—figure supplement 1A*). In addition, the levels of actin at the cortex of MCF-7 cells were increased after completion of CaAR relative to those at the outset of CaAR (*Figure 5—figure supplement 1B*).

A very sensitive readout for an increased F/G actin ratio in cells is the release of transcriptional regulators, such as the serum response factor (SRF) co-factor MRTF-A, from sequestration in the cytosol (*Miralles et al., 2003*). Indeed, we found that within a minute of $Ca^{2+}$ influx and CaAR onset, MRTF-A translocated into the nucleus in >95% of MCF-7 cells, and remained there for up to 30 min (*Figure 5A,B*). MRTF-A translocation also occurred in HeLa cells and was blocked upon INF2 knock-down (*Figure 5C*). When we blocked CaAR by treatment with LatA, MRTF-A translocation was also inhibited (*Figure 5D*). In contrast, treatment with CytoD induced CaAR-independent translocation of MRTF-A (*Figure 5E*), consistent with previous reports (*Descot et al., 2009*). We next tested whether CaAR-induced translocation of MRTF-A indeed caused SRF-mediated transcription. RNA levels of two known SRF targets, CTGF and NR4R3, were strongly increased

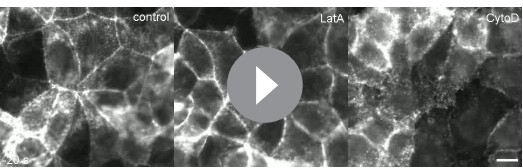

**Video 7.** MCF-7 cells expressing Lifeact-GFP, either untreated or pretreated with 400 nM LatA or 1 µM CytoD were exposed to 1 µM ionomycin at t = 0 s. Corresponds to *Figure 3A*. Scale bar: 10 µm.

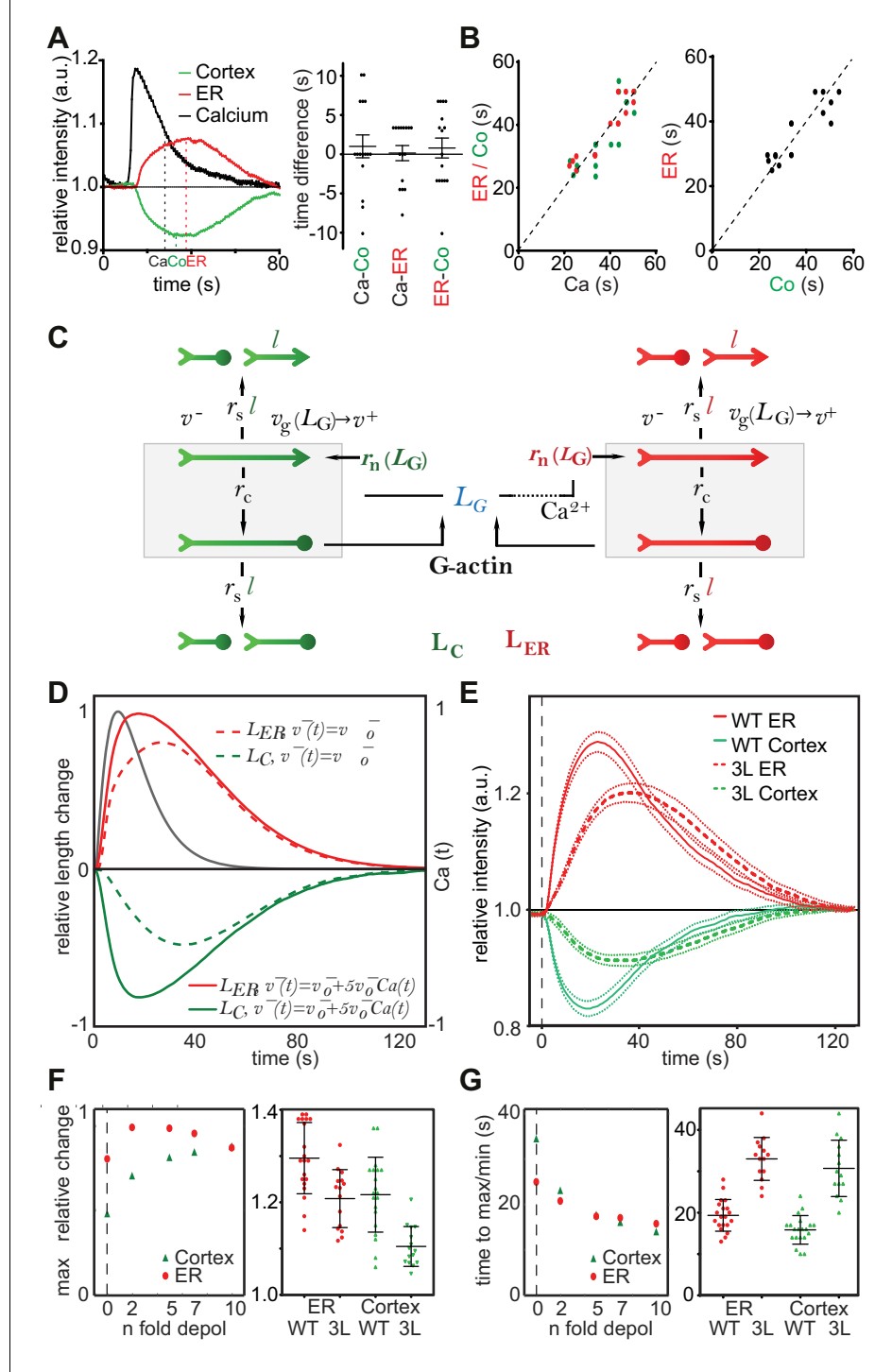

**Figure 4.** A stochastic model rationalizes CaAR features and kinetics. (**A, B**) Correlation analysis of actin and calcium dynamics in MCF-7 cells undergoing CaAR. (**A**) Temporal shift between half-maximal decay of calcium (Ca), actin maximum at nuclear periphery (ER) and actin minimum at the cell cortex (Co). (**B**) Correlation between Ca and ER (red, Pearson correlation coefficient, r = 0.924), Ca and Co (green, r = 0.840) and between ER and Co (black, r = 0.880). Dotted lines: y = x. (**C**) Stochastic model for actin competition. Filaments at the cortex (green) and ER (red) are represented by arrows (head: barbed end, circle: capped end). Total length of cortical (LC) and ER (LER) actin is indicated. Monomer pool is given by LG. Relevant parameters: Rates of severing (rs per length l), capping (rc), nucleation (rn), velocities of elongation (v+) and depolymerization (v-). See Supplementary material for details of the model. (**D**) Evolution of F-actin concentration at ER (red) and cortex (green) generated by the

*Figure 4 continued on next page*

*Figure 4 continued*

model using a simple competition scenario (dotted lines) or including calcium-activated depolymerization of actin (full lines). All simulations were run using the indicated $Ca^{2+}$ curve (black) as input. (E) Average intensity curves of Lifeact-mCherry at ER (red) and cell cortex (green) of HeLa KO cells expressing wildtype or WH2 mutant GFP-INF2-CAAX (3L: L976A, L977A, L986A). Dotted red and green lines indicate SEM. (F) Effect of different depolymerization rates on maximal actin change at cortex (green) and ER (red). Shown are predictions from simulations (left) and experimental data comparing wildtype and 3L-INF2 expressing HeLa cells (data points, mean and SD). (G) As in (F) but showing the effects of varying depolymerization rate on the time until cortical minimum or ER-maximum is reached.

The following figure supplement is available for figure 4:

**Figure supplement 1.** A stochastic model rationalizes CaAR features and kinetics.

upon CaAR induction by ionomycin, reaching a maximum at 1 hr (*Figure 5—figure supplement 1C*). We selected this time point for the analysis of CaAR-mediated changes in gene transcription. We treated MCF-7 cells with 1.5 µM ionomycin for 10 min after pre-incubation for 10 min with either LatA, CytoD or control buffer. After washout of the drugs and ionophore, we cultured the cells for another 50 min in 4-thio-uridine and collected newly synthesized mRNA (*Figure 5F*). We found that a 10 min pulse of ionomycin was sufficient to decrease the expression of 478 genes and enhance the expression of 405 genes by more than 1.5 fold (*Figure 5—source data 2*). Treatment with LatA on its own had very little effect, but upregulation of 88 ionomycin-induced genes was strongly inhibited by LatA (*Figure 5—source datas 1* and *2*). Nearly half of these genes are known SRF targets and about one third were also induced by the SRF activator CytoD (*Figure 5G*). In agreement with the reported functions of SRF, many of the identified transcriptional changes were associated with genes involved in cytoskeletal organization, calcium regulation, cell signaling and regulation of cell adhesion (*Figure 5G*). More than 60% of the CaAR-regulated genes have been associated with cancer and more than one-third have been implicated in cellular stress responses, cell

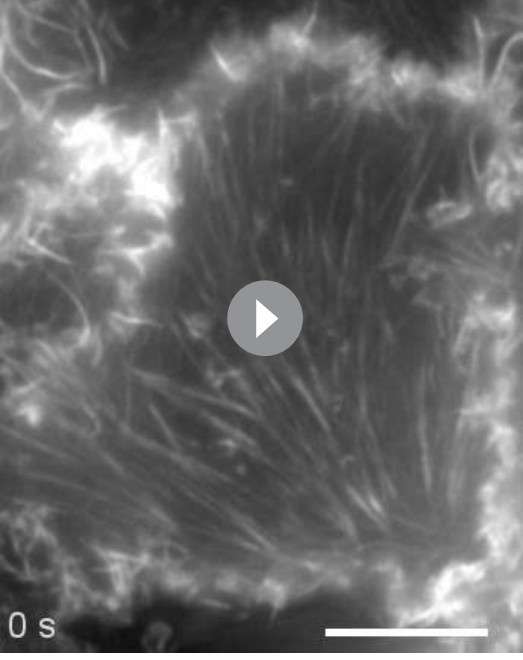

**Video 8.** Cortical actin reorganization in an MCF-7 cell expressing Lifeact-GFP stimulated by laser ablation. Corresponds to *Figure 4—figure supplement 1A*. Scale bar: 10 µm.

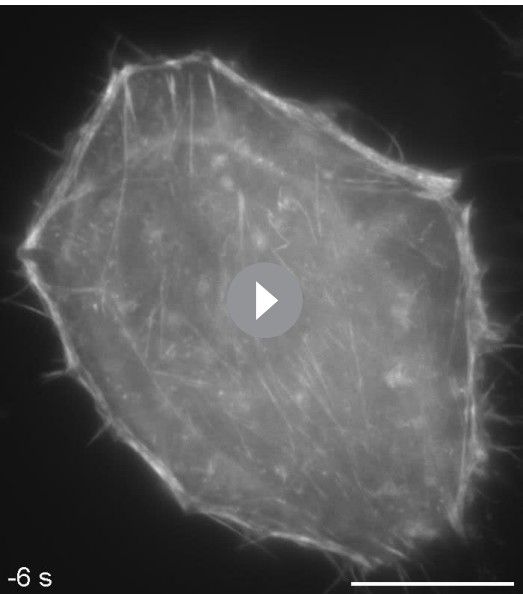

**Video 9.** HeLa cell expressing Lifeact-GFP stimulated by laser ablation. Corresponds to *Figure 6A*. Scale bar: 10 µm.

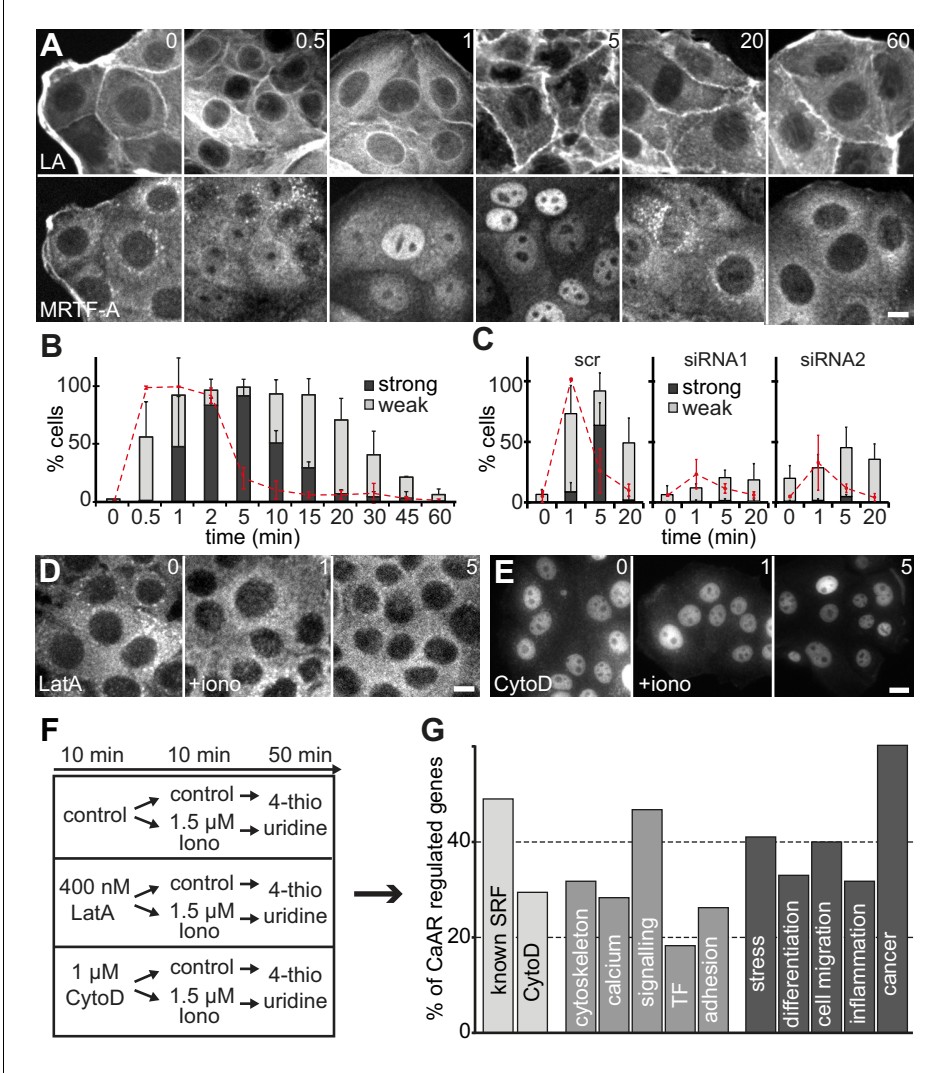

**Figure 5.** CaAR induces transcription via MRTF-A and SRF. (**A**) Immunofluorescence detection of MRTF-A in MCF-7 cells at the indicated times after addition of 1 μM ionomycin. (**B**) Quantification of MCF-7 cells exhibiting CaAR (red dotted line) and weak (light grey) or strong (dark grey) nuclear MRTF-A accumulation (mean + SD, n = 3 experiments with >100 cells per time point). (**C**) Quantification of HeLa cells exhibiting CaAR (red dotted line) and weak (light grey) or strong (dark grey) nuclear MRTF-A staining (mean + SD, n = 3 experiments with >100 cells per time point). Graphs show values for cells treated with scrambled siRNA and two INF2-siRNAs. (**D, E**) Analysis of MRTF-A localization upon stimulation of MCF-7 cells with 1 μM ionomycin (added at t = 0). Cells were pretreated for 10 min with either 400 nM latrunculin A (**D**) or 1 μM cytochalasin D (**E**). (**F**) Protocol for transcriptome analysis. (**G**) Major functional categories of CaAR-regulated genes, assembled from the manual curation of public databases and literature. Association categories are given for the known SRF regulation (light grey), cellular processes (medium grey) and biological processes (dark grey). Time in min. Scale bars 10 μm.

The following source data and figure supplement are available for figure 5:

**Source data 1.** CaAR induces transcription via MRTF-A and SRF.
**Source data 2.** Differentially regulated genes.
**Figure supplement 1.** CaAR induces transcription via MRTF-A and SRF.

differentiation, cell migration or inflammation (*Figure 5G*). SRF is known to respond to actin changes mediated through Rho GTPases (*Treisman et al., 1998*). However, we found that CaAR-mediated induction of CTGF was not affected by the ROCK inhibitor Y27632, the calcineurin inhibitor cyclosporin A (CsA) or by the CaM kinase inhibitors KN62 and KN93 (*Figure 5—figure supplement 1D*). Importantly, all CaAR transcriptome experiments were performed in serum-starved cells and addition of serum did not induce CaAR in MCF-7 cells. In addition, treatment of cells with Y27632, the Rho1 inhibitor C3 transferase or with CsA did not affect CaAR (*Figure 5—figure supplement 1E*). Our results therefore indicate that CaAR mediates SRF activation through actin reorganization, but via a $Ca^{2+}$-dependent pathway distinct from the previously described SRF activation involving Rho, ROCK and mDia (*Baarlink et al., 2013*; *Copeland and Treisman, 2002*).

## CaAR mediates acute cellular reorganization

We next wanted to understand the physiological consequences of transient actin reorganization during CaAR. When we examined the cellular consequences of CaAR in laser ablation experiments with HeLa cells, we observed accumulation of actin at the site of membrane damage in more than 90% of the cases (*Figure 6A*, *Video 9*). This accumulation was reminiscent of previously observed wound repair processes (*Clark et al., 2009*; *McNeil, 2002*) and typically occurred after completion of CaAR (*Figure 6B*). Upon knock-down of INF2, HeLa cells did not undergo CaAR and were also completely unable to recruit actin to the site of membrane wounding (*Figure 6C*). Importantly, membrane damage as measured by uptake of propidium iodide or FM4-64 was repaired within a few seconds of damage induction (not shown). Actin accumulation was therefore not required for membrane sealing but for a later step in wound repair, possibly cortex reassembly. Such repair mechanisms should be especially relevant for terminally differentiated cells that can no longer be replaced in vivo, such as podocytes (*Pavenstädt et al., 2003*). We therefore examined CaAR in *in vitro* differentiated human AB8 podocytes (*Saleem et al., 2002*). Similar to our observations in HeLa cells, laser ablation at the periphery of podocytes induced CaAR and led to subsequent actin accumulation at the wounding site, which was then efficiently sealed without apparent loss of cellular integrity (*Figure 6D*, *Video 10*). Interestingly, actin accumulation at the wound was accompanied by simultaneous induction of lamellipodia in the immediate vicinity (*Figure 6D*). In both cell types examined above, the strong accumulation of actin at cortical wounding sites occurred right after completion of CaAR. Similarly, when we extended the period of observation in MCF-7 cells that had undergone CaAR upon ATP exposure, we found that they initiated extended basal protrusions that correlated with the end CaAR. Protrusions emanating from cell-cell junctions collapsed after only 5–10 min, but those appearing at free cell edges persisted for up to 1 hr (*Figure 6E*, *Video 11*). To study this phenomenon in more detail we ablated a single MCF-7 cell in a monolayer. CaAR was efficiently induced in all surrounding cells and we again observed short-lived protrusions at cell-cell junctions and longer-lived protrusions at free cell edges (*Figure 6F*, *Video 12*). In addition, some cells formed large lamellipodia, which rapidly closed the gap left by the ablated cell (*Figure 6F*). Considering the extended activation of cell spreading and cellular protrusion after CaAR we wondered whether this would have noticeable consequences for collective migration in a typical wound healing setting. We therefore observed MCF-7 monolayers migrating into a free area after removal of a PDMS spacer (*Figure 7A*). After 12 hr, untreated cells had moved in to the gap with an average speed of 3 µm/h, while cells migrating in the presence of 50 µM ATP (and therefore exhibiting CaAR at the onset of the experiment) covered a much larger area with an average speed of 7 µm/h (*Figure 7B*). A more detailed analysis revealed that ATP treatment led to an acceleration for the initial 4 hr of migration and that cells then reverted to the speed of control cells (*Figure 7C*, *Video 13*). Our findings clearly show that ATP-mediated calcium influx and CaAR are associated with prolonged activation of protrusion and collective migration of MCF-7 cells. As we were not yet able to remove INF2 from MCF-7 cells, we cannot exclude at this time that the observed effects on protrusion could be due to a CaAR-independent effect of ATP.

In summary, our results on plasma membrane sealing and transcriptional regulation indicate that CaAR plays an important role during acute morphogenetic adaptations, with implications for such diverse processes as cell migration, cancer progression and cell stress response.

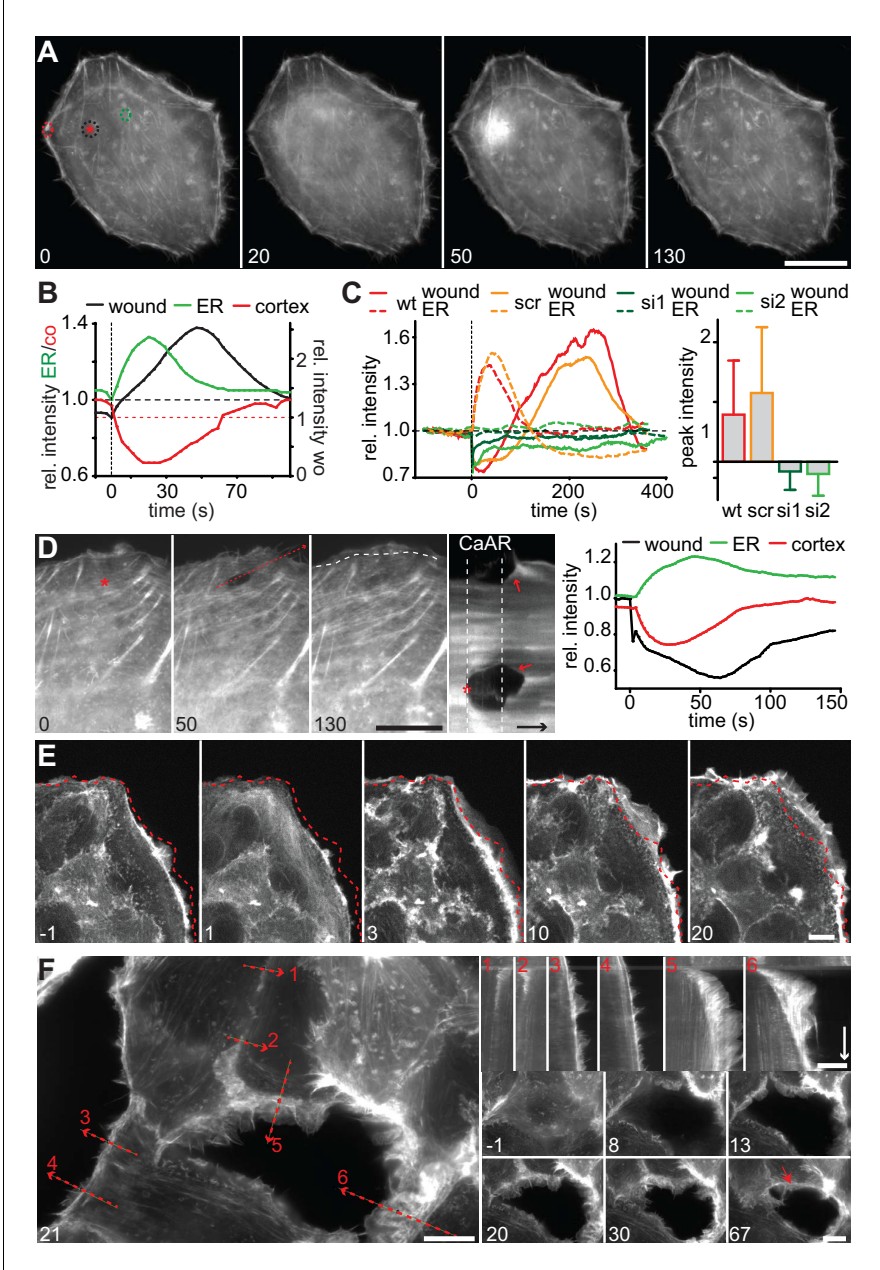

**Figure 6.** CaAR mediates acute cellular reorganization. (**A**) A HeLa cell expressing Lifeact-GFP was damaged by laser ablation and Lifeact-GFP intensity was monitored in different regions. Asterisk: ablation site. Regions for intensity measurements in (**B**) are indicated in corresponding colors. (**B**) Plots of Lifeact-GFP signal intensity at ER, cortex and at the ablation site of the cell shown in (**A**). (**C**) Plots of Lifeact-GFP signal intensity in indicated regions of control and INF2-siRNA-treated cells. Quantification of peak intensities at the wounding site is shown as mean ± SD (n > 9). (**D**) A podocyte expressing Lifeact GFP was damaged by laser ablation and subsequently monitored by fluorescence microscopy. Asterisk indicates site of laser ablation. Red dotted line indicates the path of kymograph, a white dotted line indicates the outline of the cell before ablation, and arrows indicate instances of wound-repair and lamellipodia formation after CaAR completion. Lifeact-GFP intensity curves for ER, cortex and wound site are shown in the graph. Time arrow: 50 s. (**E**) Increase in cortical actin at the cell periphery upon activation of CaAR in MCF-7 cells with 50 µM ATP. The dotted lines indicate the position of the cell boundary before stimulation. (**F**) Changes in cortical actin dynamics after ablation of an MCF-7 cell within a monolayer. Kymographs are shown along the dotted lines for indicated positions (red numbers). Time arrow: 20 min. At later time-points actin congresses into ring structures around gaps (red arrow). Times in sec (**A–D**) and min (**E, F**). Scale bars: 10 µm.

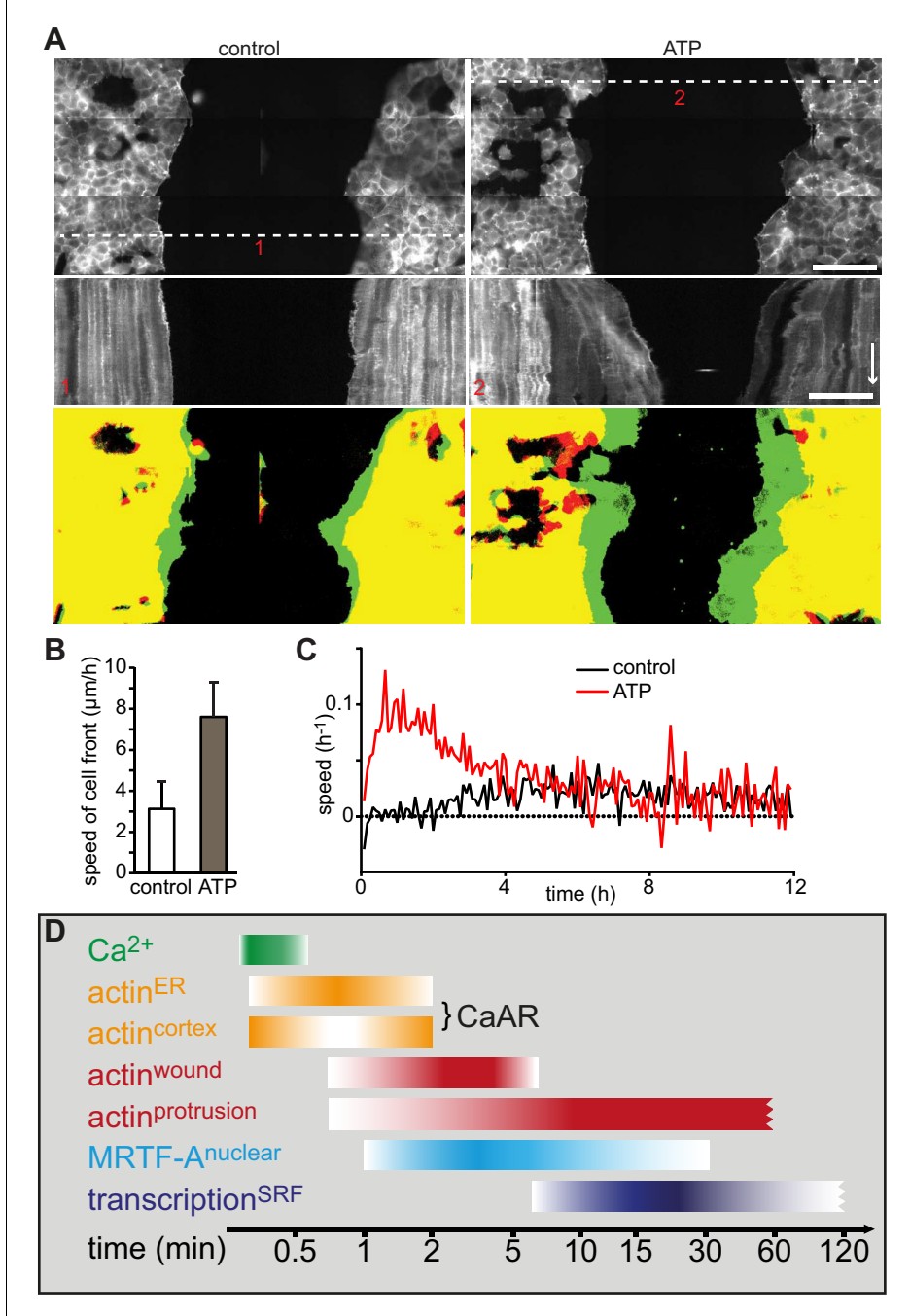

**Figure 7.** Wound healing following induction of CaAR and timeline. (**A**) Representative examples of MCF-7 cells expressing Lifeact-mCherry seeded on either side of a PDMS spacer. After removal of the spacer cells were cultured for 24 hr and then treated with either control medium (left panels) or medium containing 50 μM ATP (right panels). Shown are the cell positions at the time of the medium exchange (offset resulting from stitching), the kymographs along the indicated lines (red numbers) and the overlay between t0 (red) and t5h (green). Time arrow: 4 hr. Scale bars: 100 μm. (**B**) Rate of cell front movement for control vs. ATP-treated cells. Student's t-test $p<0.05$ for $n = 5$ experiments. (**C**) Graphs showing speed of closure in control cells vs. ATP-induced cells. (**D**) Timeline of CaAR and associated processes as discussed in the text.

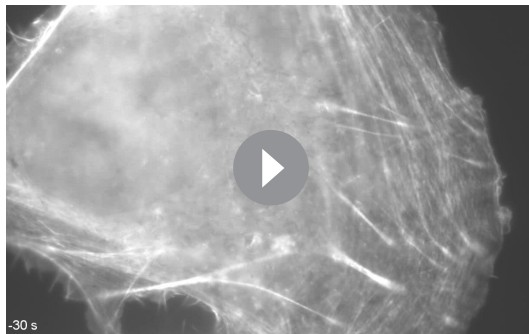

**Video 10.** AB8 podocyte expressing Lifeact-GFP stimulated by laser ablation (asterisk). Corresponds to *Figure 6D*. Scale bar: 10 μm.

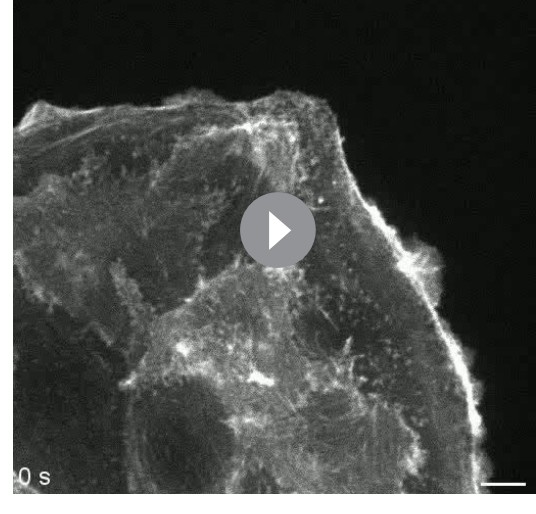

**Video 11.** Spreading of MCF-7 cells expressing Lifeact-GFP stimulated by ATP. Corresponds to *Figure 6E*. Scale bar: 10 μm.

## Discussion

Calcium has long been known to transmit acute signals in processes such as inflammation, wound healing or stress responses (*Høyer-Hansen and Jäättelä, 2007*; *Wood, 2012*). How calcium signaling is linked to cellular morphogenesis is less well understood. Interestingly, a recent study reported formation of a $Ca^{2+}$-dependent perinuclear actin ring in 3T3 fibroblasts (*Shao et al., 2015*). We have now found that increased intracellular calcium levels lead to transient and global reorganization of actin in a wide range of epithelial, mesenchymal, endothelial and hematopoietic cell lines. This reorganization can be triggered by various mechanical and biochemical signals and is characterized by simultaneous actin polymerization at the ER and disassembly of cortical actin filaments. The ubiquity, sensitivity and conserved features of the observed Calcium-mediated Actin Reset (CaAR, *Figure 7D*) indicate that this is a fundamental response of mammalian cells. In agreement with this idea, we found that CaAR can act as a morphogenetic integrator during acute cellular perturbations, such as cell cortex damage and wound healing.

The highly synchronous kinetics of actin polymerization at the ER and actin disassembly at the cell cortex strongly suggests that competition for a common pool of actin monomers occurs between actin nucleators at the two locations. Such competition between different actin structures within the same cell has recently been demonstrated for fission yeast cells (*Burke et al., 2014*; *Suarez et al., 2015*). Our stochastic model for CaAR shows that the extent and speed of actin reorganization at the cortex and ER can only be achieved if $Ca^{2+}$ simultaneously increases the rate of nucleation at the ER and the overall rate of

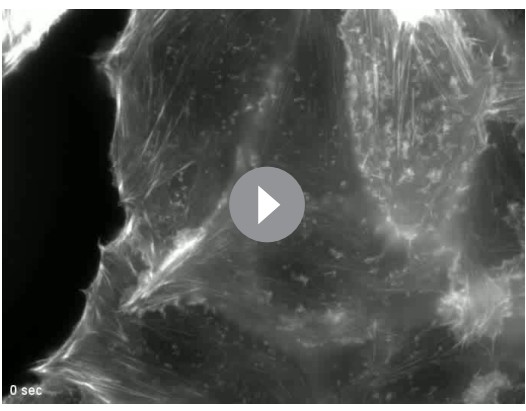

**Video 12.** MCF-7 cells expressing Lifeact-GFP stimulated by laser ablation of a single cell in a monolayer. Corresponds to *Figure 6F*. Scale bar: 10 μm.

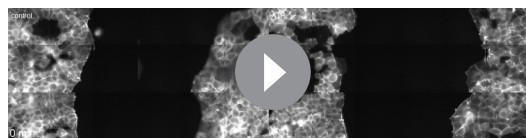

**Video 13.** MCF-7 cells expressing Lifeact-GFP migrating in to a gap in the absence (left) or presence (right) of 50 μM ATP. Corresponds to *Figure 7A*.

actin depolymerization. Interestingly, the key regulator of CaAR, INF2, is a unique molecule that can at the same time stimulate actin nucleation, elongation, severing and depolymerization (*Gurel et al., 2015*). We could validate our theoretical prediction, using a mutant INF2 with reduced depolymerization activity. Expression of this mutant in INF2 KO cells supported CaAR at much slower kinetics and reduced turnover. $Ca^{2+}$-mediated activation of INF2 could therefore account for all the experimentally observed kinetics. How then does calcium regulate INF2? We have found that INF2 binds to calmodulin, a highly abundant cellular regulator of calcium signaling. It will be interesting to test whether the calmodulin-INF2 interaction occurs directly, or via additional adaptor proteins such as IQGAP1. This protein has recently been shown to bind INF2 (*Bartolini et al., 2016*) and is a known interactor of calmodulin (*Jang et al., 2011*).

It is well established that SRF/MRTF-dependent transcription is activated by Rho GTPase and mDia-mediated actin dynamics (*Hill et al., 1995*; *Miralles et al., 2003*). We have found that CaAR provides an alternative, INF2-mediated, mechanism for MRTF activation. Interestingly, although CaAR is independent of both Rho and serum, we find a strong overlap between CaAR-regulated and classical serum-induced genes. Many of the CaAR-induced genes are associated with cytoskeletal organization, cell adhesion or signaling. This suggests a possible integration of transcriptional changes with the observed morphological effects of CaAR on actin protrusions, cortex repair and cell migration.

In summary, we have identified CaAR as a fundamental process linking $Ca^{2+}$ signaling, cell mechanics, actin dynamics and SRF-mediated transcription. Considering its basic constituents (calcium, actin), this process likely influences a multitude of signaling pathways and physiological processes. Thus, the observed consequences of CaAR should be considered when studying cellular perturbations that are linked to $Ca^{2+}$ influx. The strong association of CaAR-induced genes with cancer, inflammation and stress point to exciting opportunities for future studies of CaAR in living organisms. Of particular note is a potential role of CaAR in kidney pathology, as INF2 is known to regulate actin dynamics in podocytes, and INF2 mutations are linked to hereditary kidney diseases (*Brown et al., 2010*; *Sun et al., 2013*).

## Materials and methods

### Cell culture

Cells were grown at 37°C with 10% $CO_2$ in Dulbecco's Modified Eagle medium (DMEM-Glutamax-I; Gibco, Carlsbad, CA, USA) supplemented with 10% fetal bovine serum (FBS; Gibco). Routinely, $2 \times 10^4$ cells/ml were seeded on glass-bottomed dishes (Ibidi, Martinsried, Germany), 8-well slides (Ibidi) or 4-well LabTek dishes (Nunc, Rochester, NY, USA) and incubated for 24 hr or 48 hr before imaging. Live cell imaging was performed with cells seeded on glass bottom dishes and incubated in Hanks' buffered salt solution (HBSS) supplemented with 10 mM HEPES (pH 7.4). Apart from the experiment in *Figure 2B* labeled 'serum', all cells were starved in imaging buffer 1 hr prior to experiments. The following cell lines were used in this study: MDCK II (ECACC 0062107); MDCK II cells stably expressing Lifeact-mCherry (*Klingner et al., 2014*); MCF-7 (ECACC 86012803); MCF-7 cells stably expressing Lifeact-GFP or Lifeact-mCherry (this study); HeLa (ECACC 93021013); HeLa cells stably expressing Lifeact-GFP or Lifeact-mCherry (this study); HeLa INF2 KO cells, and HeLa INF2 KO cells stably expressing Lifeact-mCherry (this study); NIH 3T3 (ECACC 93061524); CCL-39 (ATCC-CCL-39); PANC-1 (ECACC 87092802); U-2 OS (ECACC 92022711); COS-7 (ECACC 87021302); GM7373 (DSMZ ACC109); GM7373 stably expressing Lifeact-mKate2 (*Kronlage et al., 2015*); AB8 podocytes (*Saleem et al., 2002*), AB8 cells stably expressing Lifeact-GFP (this study); HoxB8-immortalized mouse monocytes and neutrophils ([*Wang et al., 2006*] and this study); HEK293 cells (ECACC 85120602). All cell lines were checked for identity by visual inspection of morphologies and tested negative in Mycoplasma tests using the following primers: RWS2534: 5'-CGCCTGAGTAGTACG TTCGC-3'; RWS2535: 5'-CGCCTGAGTAGTACGTACGC-3'; RWS2536: 5'-TGCCTGAGTAGTACA TTCGC-3'; RWS2537: 5'-CGCCTGGGTAGTACATTCGC-3'; RWS2538: 5'-CGCCTGAGTAGTAGTC TCGC-3'; RWS2539: 5'-TGCCTGGGTAGTACATTCGC-3'; RWS2540: 5'-GCGGTGTGTACAA-GACCCGA-3'; RWS2541: 5'-GCGGTGTGTACAAAACCCGA-3'; RWS2542: 5'-GCGGTGTG TACAAACCCCGA-3'.

## Transfection and plasmids

Lifeact-acGFP, Lifeact-mCherry and Lifeact-mKate2 were expressed from pEFIRES and pGKIRES plasmid backbones. Beta-Actin (ACTB-GFP) was expressed from pEGFP-C1. Lifeact-acGFP was cloned via NotI/PacI restriction sites into the retroviral expression vector pQXCIP for stable transfection of AB8 cells. pssRFP-KDEL (*Altan-Bonnet et al., 2006*), pGFP-INF2-CAAX, pGFP-INF2-non-CAAX and pGFP-INF2(A149D)-CAAX were described previously (*Ramabhadran et al., 2011*). For expression in HEK293 cells, INF2 was first subcloned into pGADT7.3 (BspEI/XmaI-XhoI) and then into pEGFP-C3 (EcoRI-SalI). The 3L (L976A, L977A, L986A) mutants were generated by site directed mutagenesis (Stratagene Quickchange, Agilent, Santa Clara, CA, USA) using either pGFP-INF2-CAAX or pGFP-INF2-nonCAAX as a template. Primers used for mutagenesis: RWS3036: 5′-GTTCAG-CACGATGAAGGCCTTTAGGGACCTTTTCC-3′ siRNA resistant forward; RWS3037: 5′-GGAAAAGG TCCCTAAAGGCCTTCATCGTGCTGAAC-3′ siRNA resistant reverse; RWS3163: 5′-gtgtgtgtcatc-gatgccGCgGCggctgacatcaggaaggg-3′ L976A and L977A forward; RWS3164: 5′-cccttcctgatgt-cagccGCcGCggcatcgatgacacac-3′ L976A and L977A reverse; RWS3165: 5′-catcaggaagggcttccagGCgcggaagacagcccggg-3′ L986A forward; RWS3166: 5′-cccgggctgtcttccgcGCctggaagcccttcctgatg-3′ L986A reverse. All sub cloning and mutagenesis steps were verified by sequencing. Cell transfections were performed using Fugene6 or LipofectamineTM 2000 (Invitrogen) according to the manufacturer's instructions. To obtain stably transfected lines, cells were selected on 600 µg/ml hygromycin (InvivoGen, San Diego, CA, USA) and/or 600 µg/ml puromycin (InvivoGen) for 7–10 days under constant selection pressure. Antibiotics were omitted during drug treatments and imaging. For stable retroviral transduction of AB8 cells, GP2-293 cells were transfected using the calcium phosphate method with 5 µg pQXCIP-Lifeact-acGFP and 5 µg pVSV-G (Clontech, Mountain View, CA, USA) in a 10 cm dish. After 6 hr, the medium was replaced and cells were grown for another 72 hr. Virus-containing supernatant was filtered through a 0.45 nm filter (Millipore) and polybrene (8 µg/ml) was added. AB8 cells in a 6-well dish were transfected with 2 ml of the virus-containing medium and 2 ml of fresh AB8 medium. After 24 hr, the medium was replaced and cells were allowed to recover for 24 hr. Transduced cells were selected with puromycin (2 µg/ml) (*Schulze et al., 2014*; *Wennmann et al., 2014*).

## Fluorescence microscopy

Epifluorescence imaging was performed on a fully automated iMIC-based microscope from FEI/Till Photonics, using an Olympus 100 × 1.4 NA objective and DPSS lasers at 488 nm (Cobolt Calypso, 75 mW) and 561 nm (Cobolt Jive, 150 mW) as light sources. Lasers were selected through an AOTF and directed through a broadband fiber to the microscope. A galvanometer-driven two-axis scan head was used to adjust laser incidence angles. Images were collected using an Imago-QE Sensicam camera. Acquisition was controlled by LiveAcquisition software (Till Photonics). Fluorescence Recovery After Photobleaching (FRAP) of actin-GFP was performed using a third galvanometer-controlled mirror (Polytrope) to switch between wide-field and FRAP modalities. Ablation experiments were carried out on an iMIC setup equipped with a pulsed 355 nm picosecond UV laser (Sepia, Pico-Quant) as previously described (*Raabe et al., 2009*). Confocal microscopy was performed on an iMIC42 setup equipped with a spinning disk unit (Andromeda) using Olympus 20x air (NA 0.75) and 60x oil immersion (NA 1.49) objectives. Images were taken using typical filter settings for excitation and emission of fluorescence probes/proteins and recorded on EMCCD cameras (Andor iXon Ultra 897).

## Drug treatment

Cells were treated with 50 µM blebbistatin (Sigma-Aldrich, St Louis, MO USA) to inhibit myosin II ATPase activity, 400 nM latrunculin A (Enzo Life Sciences) to sequester actin monomers, 1 µM thapsigargin to inhibit $Ca^{2+}$ uptake into the ER, 1 µM cytochalasin D (Sigma) to depolymerize actin, 10–50 µM cyclosporin A (Sigma) to inhibit calcineurin, 1–10 µM KN93 or KN62 (Tocris) to inhibit $Ca^{2+}$/cal-modulin-dependent protein kinase II, 10–20 µM Y27632 (Sigma) to inhibit ROCK, 2.5 µg/µl C3 transferase (Cytoskeleton) to inhibit Rho1, 1 µM nocodazole (Sigma) to depolymerize microtubules, 10–20 µM SMIFH2 (Sigma) to inhibit formins and 50–100 µM CK666 or CK869 (Sigma) to inhibit the ARP2/3 complex.

## siRNA experiments

Silencer Select (21 nt) siRNAs were purchased from Ambion/Lifetechnologies: siRNA1 (s34736): sense sequence 5'-CCAUGAAGGCUUUCCGGGAtt-3'); siRNA2 (s230622): sense sequence 5'-GCA UUGUCAUGAACGAGCUtt-3'. As a negative control, a random, non-targeting siRNA sequence was used. HeLa cells stably expressing Lifeact-GFP were seeded on coverslips or in 8-well Ibidi slides and transfected with siRNAs (30 nM) on the next day, using Oligofectamine (Invitrogen, Carlsbad, CA, USA) according to the manufacturer's instructions. Cells were incubated for 72 hr and either imaged live for CaAR after ionomycin addition (*Figure 3E*, *Figure 3—figure supplement 1C*) or laser ablation (*Figure 3—figure supplement 1D* , *6C*) or fixed at various time-points after ionomycin addition and immunostained for MRTF-A (*Figure 5C*).

## Generation of INF2 knock out cells

To knock out INF2 in human cell lines we obtained a mix of three different CRSPR/Cas9 plasmids and the corresponding HDR plasmids from Santa Cruz (sc-410096). gRNA sequences: A: Sense: GAGGAGCTGCTGCGAGTCTC; B: Sense: GGTCGACATGAGCAGCCACC; C: Sense: CAGCGA-CAACGTGCCCTACG. HeLa wt cells were co-transfected with both plasmid mixes using Fugene6 and grown for 24 hr without selection. We visually confirmed the appearance of RFP expressing cells indicated successful disruption of INF2. We next grew cells for two weeks under selection pressure with 0.5 µg/ml puromycin. The population of stable puromycin resistant cells was then transfected twice with a Cre expression vector (Santa Cruz) using Fugene6 to remove the RFP and puromycin cassette. Finally, we selected individual INF2 KO clones by limited dilution in 96-well plates. All clones were characterized for INF2 expression by Western blot and immunofluorescence and for absence of CaAR induction (Rhodamine-phalloidin staining) upon ionomycin stimulation.

## Shear flow experiments

For shear-flow experiments, $5 \times 10^3$ cells were seeded in µ-slide$^{0.2}$ Luer flow chambers (Ibidi), incubated for 48 hr and then connected to the Ibidi pump system, perfused with DMEM and subjected to 20 dyn/cm$^2$ oscillatory shear stress at 0.2 Hz.

## Atomic force microscopy (AFM)

For elasticity measurements and mechanical perturbation experiments (AFM), cells were seeded (at $2–5 \times 10^5$ cells/ml) on 35 mm Fluorodish glass-bottomed dishes (WPI) 48 hr prior to experiments. All experiments were performed using a Nanowizard III AFM (JPK Instruments, Berlin, Germany) integrated into a TCS SP8 confocal laser scanning microscope (Leica, Wetzlar, Germany). Gold-coated MSCT cantilevers with spherical (10 µm) polystyrene probes (Novascan, Ames, IA; USA) with a nominal spring constant of 0.01 pN/nm were used to quantify cell elasticity. Force-distance curves were acquired with a z-length of 2.5 µm at a tip velocity of 1 µm/s and a retracted delay of 1 s (~1 force distance curve per 6 s). At a loading force of 0.75 nN the cantilever reached a maximal indentation of 1 µm (*Carl and Schillers, 2008*). Elasticity was measured as Young's modulus based on Sneddon's model of nanoindentation using Protein Unfolding and Nano-Indentation Software (PUNIAS, http://punias.voila.net). The last 200 nm of the force distance curve was analyzed to quantify the elasticity of the cell cytoplasm. All elasticity measurements were performed at RT. For all cell perforation experiments, Multi75-G Cantilevers (BudgetSensors, Sofia, Bulgaria) with spring constants ranging from 2.5 to 3.5 N/m were used. Cells were stimulated by placing the cantilever tip in the center of the nuclear area and indenting them with a speed of 1 µm/s. To determine the force threshold for CaAR induction, maximum loading forces ranging from 5 nN to 50 nN were used. To study repeated CaAR induction, cells were indented ten times with a maximum loading force of 50 nN and variable pauses between single indentations. Fluorescence images were acquired with a 63x HC PL APO CS2 oil immersion objective (NA = 1.4) and hybrid detection system for photon counting (Leica HD™). Z-stacks of cells (distance 500 nm) were taken every 30 s. All confocal fluorescence images were analyzed and processed using Fiji. Fluorescence intensity was measured in unprocessed images. For presentation, grouped z-projections of three slices were cropped and contrast-adjusted.

## Image processing and analysis

Images were processed using Fiji and Matlab (Mathworks Inc., Natrick, MA). Custom made Matlab scripts are included as Figure2-source codes 1–3. Images were contrast-adjusted and zoomed for purposes of presentation in figures only. For image cleanup and denoising we routinely used the background subtraction algorithm in Fiji (radius 50 pixel). Kymographs and intensity plots were created using the respective features in Fiji.

## Calcium imaging

Cells were loaded with Fluo-4 (5 µM, Thermo Fisher) or Fura-2 (3 µM, Thermo Fisher) for 15–30 min at 37°C. Fluo4 was excited at 488 nm. Fura-2 intensity was determined by ratiometric measurement using excitation at 340 and 380 nm (detection at 500 nm). Fura-2 fluorescence was acquired using an Axiovert 200 (Zeiss, Wetzlar, Germany) equipped with a VisiChrome high speed polychromator system (Visitron System, Puchheim, Germany), a CoolSNAP fx camera (Photometrix) and Metafluor imaging software (Visitron System). $Ca^{2+}$-dependent fluorescence was acquired every 10 s in alternation with imaging of Lifeact-GFP fluorescence.

## Immunofluorescence and cell labeling

Cells were grown on glass coverslips, fixed with 4% paraformaldehyde in PBS for 20 min, washed with PBS and permeabilized with 0.1% Triton X-100 for 10 min prior to incubation with primary Ab for 1 hr. After incubation with secondary antibodies and/or Rhodamine-phalloidin (Invitrogen, R415) for 1 hr in PBS, cells were washed and subsequently mounted on slides in Mowiol/Dabco (Roth, 0713 and 0718). Primary antibodies: rabbit anti-INF2 (*Chhabra et al., 2009*), goat anti-MRTF-A (SantaCruz: C-19; sc-21558, RRID:AB_2142498). Secondary antibodies: Alexa-Fluor 568 goat anti-rabbit/mouse, Alexa-Fluor donkey anti-goat 568 and Alexa-Fluor donkey anti-rabbit 647 (all Invitrogen). LysoTracker Red (Thermo, L7528) was used to label lysosomes. MitoTracker Red CMXRos (Thermo, M7512) was used to label mitochondria.

## Western blot analysis

For detection of INF2 and actin in Western blots, equal amounts of cell lysates were separated by SDS-PAGE, transferred to Immobilon™-P-membrane (Serva, 42581.01), incubated in primary Ab (rabbit anti-Inf2, Proteintec: 20466–1-AP, RRID:AB_10694821) and (*Chhabra et al., 2009*), (mouse monoclonal anti-β-actin, Abcam: ab125248, RRID:AB_11140352), in the presence of 5% skim milk in TBS-T overnight and labeled with HRP-coupled secondary Ab for one hour. The signal was detected using ECL chemiluminescence on an Intas Imager (Intas, Göttingen).

## Coprecipitation of INF2 with calmodulin

Cell culture, transfection, coprecipitation with calmodulin and detection of associated proteins were conducted as described previously (*Hou et al., 2015* PLoS Biol.). Briefly, HEK293 cells were cultured at 37°C under 5% CO2 in DMEM medium (Gibco Life technologies) additionally supplied with 10% fetal bovine serum, 0.1 mM gentamicin and 1x penicillin (100 units/ml)-streptomycin (100 µg/ml). GFP and GFP-INF2-CAAX were transfected into HEK293 cells using TurboFect (Thermo Scientific). 48 hr after transfection, cells were lysed in EGTA-free lysis buffer (10 mM HEPES, 1% (v/v) Triton X-100, 100 mM NaCl, 0.1 mM MgCl2, pH 7.5) supplied with 1x EDTA- free protease inhibitor cocktail and 200 µM calpain inhibitor I (Sigma) for 30 min and the supernatants containing soluble proteins were obtained after centrifugation (20.000 g, 20 min at 4°C). Cell lysates were supplied with 1 mM EGTA and 500 µM $Ca^{2+}$, respectively, then incubated with pre-washed 25 µl calmodulin-sepharose 4B (GE Healthcare) for 3 hr. calmodulin bound proteins were washed three times then eluted and denatured in 15 µl 8 M Urea and 15 µl 4x SDS-sample buffer by boiling for 5 min at 100°C. Protein samples were separated by SDS-PAGE using 9.5% acrylamide separation gels followed by western blot detection using anti-GFP antibodies (Abcam, Ab290). Results are representative of at least three experiments.

## Measurement of F/G actin ratio

MCF-7 cells were serum starved for 1 hr in HBSS and then treated with 50 µM ATP for 0.5, 1, 2 or 5 min. Cells were scraped off in ice-cold actin stabilization buffer (50 mM PIPES – pH9, 50 mM NaCl, 5

mM MgCl$_2$, 5 mM EGTA, 5% glycerol, 0.1% NP-40, 0.1% Triton-X-100, 0.1%, Tween 20, 0.1% 2-mercapto-ethanol, 0.001% antifoam C, 0.004 mM TAME, 15 μM leupeptin, 10 μM pepstatin A, 10 mM benzamidine). Cleared supernatants were centrifuged at 100,000 g for 1 hr at 4℃. High speed supernatant containing G-actin was recovered, and the pellet containing F-actin was re-solubilized with ice cold actin stabilization buffer. The F/G-actin ratio was determined by scanning densitometry using ImageJ software.

## RT-PCR

For analysis of CTGF and NR4A3 mRNA levels, total RNA was extracted from $1 \times 10^6$ MCF-7 cells using the RNeasy Mini Kit (Qiagen) following the manufacturer's protocol. cDNA synthesis was performed from 2 μg total RNA using cDNA Reverse Transcription Kit (Thermo Fisher). QPCR was performed in triplicates from 3–4 biological replicates. QPCR primers are available upon request. mRNA levels were normalized to those of B2M and GAPDH (*Figure 5—figure supplement 1C,D*). Treatment with drugs was performed at the indicated concentrations. Drugs were added for 10 min before stimulation with ionomycin.

## Transcriptome analysis

MCF-7 cells were starved in HBSS buffer for 1 hr before stimulation with 1.5 μM ionomycin for 10 min. After ionomycin washout, cells were incubated for an additional 50 min in the presence of 4-thiouridine. Cells were then lysed and total RNA was extracted from $1 \times 10^7$ MCF-7 cells using the RNeasy Mini Kit (Qiagen) following the manufacturer's protocol. LatA (400 nM) or CytoD (2 μM) were added 10 min before ionomycin and removed together with ionomycin. 4-thiouridine-labeled RNA was chemically biotinylated (Biotin HPDP, Thermo Fisher) and purified using streptavidin-coated magnetic beads (μMACS, Miltenyi). Subsequent labeling of samples for array analysis was performed with the GeneChip 3'IVT labeling assay (Affymetrix). Samples were hybridized to the GeneChip Human Gene 2.0 ST Array following the instructions from the supplier (Affymetrix). Quality control and array processing was done using GCRMA (*Wu and Irizarry, 2004*) for expression and LIMMA (*Smyth and Speed, 2003*) for elementary array comparisons and computation of fold-changes and p-values.

## Statistics

Mean values, standard deviation (SD and number of measurements (n) are provided for quantified results. Values were always pooled from at least three independent experiments. All replicates are biological replicates. Error bars in graphs are explained in the respective legends. Statistical comparison between conditions was performed using the unpaired t-test with Welch correction for single comparisons or ANOVA with Bonferroni post-test for multiple comparisons. For transcriptome analysis, genes were considered differentially regulated if expression levels deviated >1.5 fold from the mean of control cells and p value (ANOVA) was <0.05 (*Figure 5—source data 2*).

## Stochastic model for actin competition

Our initial hypothesis was that, after stimulation, a calcium-regulated polymerization activity at the endoplasmic reticulum (ER) begins to compete with actin polymerization at the cortex for a limited amount of actin monomers. We therefore considered two populations of polymerized actin (F-actin) coupled to a common pool of actin monomers (G-actin, *Figure 4C*). We assumed that the available G-actin, characterized by the equivalent length $L_G$ of actin into which it can potentially polymerize, is homogeneously distributed in the cytoplasm. F-actin populations at the cortex and ER were modeled as ensembles of individual filaments. The state of these populations is characterized by their respective total lengths of polymerized actin, $L_C$ and $L_{ER}$. Initially, we consider the cortical population in steady-state equilibrium with the G-actin pool. Nucleation at the ER is then activated and regulated by the calcium influx signal. The ER-mediated decrease in monomer availability in turn reduces polymerization at the cortex. Ongoing actin turnover will then drive cortical actin disassembly. When the calcium signal has dissipated, nucleation at the ER stops and ER actin disassembles. This restores the G-actin pool, allowing cortical actin to recover again.

## Simulations of filament dynamics

Each actin filament is characterized by its length $l$. Filaments grow at one end with a pool-dependent growth velocity $v_g(L_G)$ and shrink at the other end with a constant velocity $v^-$, leading to a net growth velocity $v^+(L_G) = v_g(L_G) - v^-$. The model also considers the interaction of actin with capping and severing proteins. Growing barbed ends are capped at a constant rate $r_c$. Since the average uncapping time is much longer than the lifetime of a filament in the cell (*Pollard et al., 2000*), we assumed that a capped filament depolymerizes from its shrinking pointed end until it is completely disassembled. We allow severing of a filament uniformly along its length, with a rate $r_s$ per unit of length. We assumed that the severing protein immediately caps the resulting barbed end, so that the lagging actin filament is always shrinking. The state of the leading fragment remains unaltered (*Figure 4C*). New filaments are nucleated in a growing state at a rate that is dependent on the size of the G-actin pool $r_n(L_G)$. Actin is therefore drained from the monomer pool by growing filaments of each population and is recycled back by the shrinking capped filaments. We assumed that depolymerized G-actin is immediately available for polymerization again, neglecting the dynamics of ADP to ATP nucleotide exchange. Neither filament elongation, nor protein activities are diffusion limited. Finally, we neglected filament annealing.

We considered that the size of the G-actin pool affects only the polymerization speed of the growing barbed end and the nucleation rate within each population. We modeled these dependencies as sigmoid-type Hill functions. For the case of the growth velocity, we consider a relation with Hill coefficient unity:

$$v_g(L_G) = v_g^M \frac{L_G}{L_G + L_s}$$

where $v_g^M$ is the saturation constant. For the nucleation rate we choose the following form

$$r_n^X(L_G) = r_{n,M}^X \frac{L_G^{h^X}}{L_G^{h^X} + L_s^{h^X}}$$

where $X$ stands for either the cortical (*C*) or the endoplasmic reticulum (*ER*) F-actin populations. Both the saturation rate $r_{n,M}^X$ and the Hill coefficient $h^X$ are chosen independently for each population. These relations imply that, as the G-actin pool size tends to zero, both polymerization and nucleation of actin are suppressed. On the other hand, for $L_G \to \infty$, the dynamic rates saturate to a maximum value and the available G-actin is no longer a limiting factor. For simplicity, we assume a common cross-over length $L_s$, at which the rates achieve their half-maximum value, and which marks the boundary between the strongly pool-limited domain and the saturated domain.

## Competition model

We initially assumed that cortical and the ER populations only differ by their nucleation mechanisms. All other interactions within each population occur with the same background rates. We considered a network type of nucleation at the cortex with a linear dependence of the nucleation rate on the amount of available G-actin, i.e.

$$r_n^C(L_G) = r_{n,M}^C \frac{L_G}{L_G + L_s}$$

The stronger nucleation activity at the ER was modeled with a second-order dependence on the size of the monomer pool. We additionally introduce the experimentally observed calcium signal as a time-dependent modulating mechanism for ER nucleation. Based on our experimental data, we modeled the calcium time-dependent signal as a normalized two-parameter Gamma distribution. We fixed the position of its maximum to the observed $t_{\max}$ and chose the shape parameter $\alpha$ to optimally fit the data. The calcium signal is given by:

$$Ca(t) = \left[ \frac{t}{t_{\max}} e^{1 - \frac{t}{t_{\max}}} \right]^{\alpha - 1}$$

where $t_{\max} = 8.4$ s and the decay is best fitted with $\alpha = 2.16$. The ER nucleation rate is then:

$$r_n^{ER}(L_G, t) = r_{n,M}^{ER} \frac{L_G^2(t)}{L_G^2(t) + L_s^2} Ca(t)$$

## Simulations and model parameters

We implemented our model using a standard Brownian dynamics approach. Each actin filament is a particle characterized by its length, lifetime and state (growing or shrinking). Capping, severing and nucleation are stochastic events, while growth and shrinking are deterministic events. The program tracks the time evolution of the respective population lengths in the coupled cortex / G-actin / ER system. The total available actin in the system $L_T$ is fixed and it is set to be completely monomeric at the start of the simulation. Using the selected model rates, the cortical population is then allowed to evolve until it is in equilibrium with the G-actin pool. In this pre-stimulus steady state, a calcium signal is then initiated at time $t = 0$, which activates actin nucleation at the ER and drives the system out of equilibrium. The simulation then runs until the cortical actin again reaches a steady state after the ER population has disappeared. *Table 2* summarizes the program input parameters. The values shown are either taken from the literature or estimated to simulate a physiologically relevant system. It is important to note that we only specify the saturation maxima of nucleation rates and polymerization velocity. The actual simulation values varied as a function of $L_T$ to determine the pre-stimulus steady state of the cortical population. The growth maximum $v_g^M$ was chosen such that even profilin-accelerated elongation rates (*Pollard, 1986*) could be accounted for when considering cellular G-actin concentrations (*Pollard et al., 2000*) and filament geometry (*dos Remedios et al., 2003*). The nucleation rate for the cortical population was estimated to obtain the amounts of polymerized actin in the range of experimental observations. Approximate capping and severing rates were taken from the literature. The cross-over length scale $L_s$ was freely chosen and total available actin in the system $L_T$ was given in $L_s$ units.

## Pre-stimulus steady state

We first considered a single cortical actin population in equilibrium with the monomer pool. Using a mean-field description for this pre-stimulus system, a set of equations can be written describing the evolution of filament length density. Using the steady-state solution of this mean-field model, we find expressions for observables, such as filament number, actin turnover and cortical length, as a function of the steady-state size of the monomer pool. It can be shown that the total actin length for this system is conserved, which leads to a self-consistent equation for the steady-state G-actin length. With this solution, analytical observables for given model parameters can be compared to time-averaged values from simulations. We used this analytical model to validate our stochastic simulations. We studied the behavior of the steady-state as a function of different model parameters. In particular, we looked at the turnover time and the ratio of polymerized to monomeric (F/G) actin. The turnover time reflects how dynamic the cortical actin is and how fast it can reorganize. Since the amount of actin is conserved in the system, the steady-state F/G actin ratio reflects how limiting the monomer pool is when a competing F-actin population emerges.

**Table 2.** Model Input parameters.

| Parameter | Symbol | Value | Reference |
|---|---|---|---|
| maximum polymerization speed | $v_g^M$ | 15.6 µm s$^{-1}$ | (*Blanchoin et al., 2014*) |
| depolymerization speed | $v^-$ | 0.1 µm s$^{-1}$ | (*Mogilner and Edelstein-Keshet, 2002*) |
| capping rate | $r_c$ | 3 s$^{-1}$ | (*Pollard et al., 2000*) |
| severing rate | $r_s$ | 0.005 µm$^{-1}$s$^{-1}$ | (*Gurel et al., 2014*) |
| maximum cortical nucleation rate | $r_{n,M}^C$ | 100 s$^{-1}$ | estimated |
| maximum ER nucleation rate | $r_{n,M}^{ER}$ | [500–1100] s$^{-1}$ | estimated |
| length scale | $L_s$ | 2000 µm | assumption |
| total available actin | $L_T$ | 4.0 $L_s$ | estimated |

## The effect of total available actin

We first examined the effect of the total amount of actin in the system, $L_T$, on the pre-stimulus steady state, given the cortical parameters shown in *Table 2*. The simulation results for the turnover time and the F/G ratio for $L_T$ = [0.5–50.0] $L_s$ are shown in *Figure 4—figure supplement 1D*. If the available actin is limited, actin polymerization cannot overcome depolymerization, resulting in a low F/G ratio. For very large G-actin pools, nucleation rate and growth velocity approach constant values. The total cortical length therefore has an upper limit and surplus actin monomers only serve to decrease the F/G ratio. We found that there is a given amount of actin for which the fraction of polymerized actin in equilibrium is maximal, given a defined set of parameters. This maximal F/G ratio corresponds to the system with the smallest monomer pool and potentially with the fastest competition upon coupling with a second actin population at the ER. We therefore chose this case, with $L_T$ = 4 $L_s$, to study the full model. The turnover time monotonically increases with $L_T$, with a saturation value corresponding to the saturation of growth speed. In our model the nucleation rate does not affect actin turnover. In steady state, the turnover time is equivalent to the time required for full polymerization of F-actin. The cortical half-time for the system with maximal F/G ratio is 12 s, which is in good agreement with half-times observed at the cortex for MCF-7 cells. This corresponds to a pre-stimulus system with 60% of actin in a polymerized state, which is consistent with published observations (*Suarez et al., 2015*).

## Actin turnover

We next studied the influence of filament depolymerization, capping and severing on cortical actin turnover. We independently varied the respective parameter values in *Table 2*. Given that disassembly refills the G-actin pool, which in turn increases polymerization speed, the behavior of the turnover time is not trivial. If we assume that turnover changes monotonically in the studied parameter interval, we can estimate how strongly actin turnover depends on each rate in the physiologically relevant parameter range.

In order to simplify the comparison, we represent the varied parameter as $r_\sigma$ and its control rate in *Table 2* as $r_\sigma^o$. Each rate was changed by at least one order of magnitude while the others were kept fixed. The results are shown in *Figure 4—figure supplement 1C*. Changing depolymerization speed has the strongest effect on actin turnover. Capping and severing in our model also affect disassembly but to a much lesser degree. This confirms the view that actin turnover is primarily driven by filament depolymerization (*Blanchoin et al., 2014*; *Mogilner and Edelstein-Keshet, 2002*) but tuned by capping (*Miyoshi and Watanabe, 2013*) or severing.

## Competition between two actin populations

In the following section we consider coupling of a second F-actin population to the pre-stimulus cortex / G-actin system introduced above. We studied the temporal evolution of the total lengths of cortical and ER populations, normalized to the pre-stimulus cortical F-actin length. The transition is monitored from the start of the calcium peak until complete depolymerization of the ER population (*Figure 4D*). The normalized calcium curve is shown in gray. We characterized the transition by looking at the maximum ER / cortex length changes, $\Delta L_{ER}$ / $\Delta L_C$, and the times at which these maxima occur, $t_{max}^{ER}$ / $t_{min}^C$.

## Competition only

We found that the simulated calcium regulation was enough to generate transient actin reorganization from the cortical population to the ER (*Figure 4D*). We next attempted to fit our free parameters to achieve a reduction in cortical actin below 50% of the pre-stimulus state and cause both populations to change synchronously, in accordance with the experimental observations (*Figure 4D–F*). Following our hypothesis that competition for a common G-actin pool drives the transition, we attempted to increase competition by varying the nucleation activity at the ER and by changing the initial monomer availability in the pre-stimulus state. We set the ER nucleation rate at the peak calcium concentration to 5–11 times the cortical nucleation rate. For each of these nucleation rates, we additionally varied the amount of total available actin in the system. The results for the maximum length changes at ER and cortex are shown in *Figure 4—figure supplement 1E* (upper panel). The rates of ER nucleation used (from left to right in each group) were $r_{n,M}^{ER}$ = [500, 700, 900, 1100] s$^{-1}$.

The amount of available actin was $L_T$ = [0.5, 2, 4, 6, 8, 12] $L_s$. We found that while increasing nucleation at the ER or available actin was able to augment the size of the ER population, it had hardly any effect on cortical actin. In the optimal case, cortical actin was only decreased to half its initial value. We were also unable to reduce the delay between the decrease in cortical actin and the increase in polymerization at the ER given as $t_{min}^C - t_{max}^{ER}$ (*Figure 4—figure supplement 1E*, lower panel). In summary, a mechanism based on simple competition between two polymerization activities does not account for the observed features of CaAR.

## Model with accelerated turnover

In order to increase the rate of the cortical reaction during CaAR, we considered an additional mechanism driven by the calcium signal, which accelerates actin turnover in the cell. Based on our results for a single actin population, we tested the effects of modulating either filament depolymerization speed $v^-$ or severing rate $r_s$ via calcium. We described time-dependent shrinking of capped actin filaments as:

$$v^-(t) = v_o^- + v_{acc}^- \, Ca(t)$$

where $v_o^-$ is the constant depolymerization speed considered before (*Table 2*) and $v_{acc}^-$ is an extra parameter representing the maximum effect of a calcium-modulated depolymerization mechanism. We consider increased depolymerization to affect cortical and ER actin in equal measure. Similarly, we consider an increased severing rate per unit length as:

$$r_s(t) = r_s^o + r_s^{acc} \, Ca(t)$$

We compare the individual effects of inducing an overall acceleration either in the depolymerization or in the severing activity. In both cases, we studied the effect of accelerating parameters on the maximal cortex decrease and its timing relative to the ER increase (*Figure 4E*). The vertical line in the plots corresponds to the pure competition model without a direct effect on the cortical dynamics. The values used for the acceleration in depolymerization were $v_{acc}^-$= [0.2, 0.5, 0.7, 1.0] μm s$^{-1}$. The values used for the acceleration in the severing rate were $r_s^{acc}$= [0.01, 0.025, 0.035, 0.05] μm$^{-1}$ s$^{-1}$.

Relative to increasing the severing activity, increasing the turnover by filament end depolymerization leads to a higher cortical depletion and a more synchronized transition between the ER and the cortical actin. Inducing a stimulus-dependent decrease in the nucleation rate at the cortex leads to a slow and weak effect on the cortical actin, so we confirm that the cortex reaction must be driven by a depolymerization mechanism.

We compared our previous results for the cortical length decrease in all runs to those obtained using our model with accelerated depolymerization. As expected, the cortical actin length does indeed show a more pronounced decrease in comparison to the results for the pool-size competition only (*Figure 4—figure supplement 1E* middle panel). The greater synchronization of the transition for all simulated parameters (*Figure 4—figure supplement 1E* lower panel) provides further confirmation of the proposed mechanism. Even if competition for a limited pool of G-actin does have an impact on the transition, as seen in the small variations across runs, the strongest influence on the cortical reaction comes from the enhancement of the overall depolymerization velocity. It should be noted that, although this has only a slight effect on ER actin assembly, a high nucleation activity ensures that rates of actin polymerization at the ER remain high. Moreover, the high turnover in the cell keeps high levels of G-actin available for polymerization, which screens the disassembly effect at the ER as long as the calcium signal is sufficiently strong.

## Simultaneous increase in severing and depolymerization

We further considered a calcium-mediated increase in the turnover due to a combined effect of enhanced filament depolymerization and a higher severing rate. In other words, we consider both $v^-$ and $r_s$ to be time-dependent.

Previously, we showed that a five-fold increase in the depolymerization results in rapid cortical disassembly in synchrony with polymerization at the ER (*Figure 4D*). Given the moderate, but still appreciable, the influence of severing on the turnover, we tested if a higher rate of severing can compensate for a weaker acceleration of depolymerization. We studied extra depolymerization

terms given by: $v_{acc}^- =$ [0.05, 0.1, 0.2, 0.3] µm s$^{-1}$. For each of these values, the extra maximum severing was varied as $r_s^{acc} =$ [0.01, 0.025, 0.035, 0.05] µm$^{-1}$ s$^{-1}$, corresponding to a 2- to 10-fold increase in the background rate of severing.

We compared the results obtained with an independent calcium-regulated change in the severing with those for a combination of both disassembly mechanisms (*Figure 4F*). Introducing additional depolymerization acceleration clearly leads to a stronger and faster cortical decrease for any severing acceleration, as expected from the strong dependence of turnover on this parameter. This effect is greater for lower depolymerization since severing is length dependent. For a given value of $v_{acc}^-$, extra severing also promotes higher cortical disassembly and reduces the delay with respect to ER actin polymerization. Thus the effect of a five-fold increase in the depolymerization rate can also be achieved by combining a three-fold increase in depolymerization velocity with a seven-fold increase in the severing rate. The increase in depolymerization velocity could be further reduced to two-fold if the severing rate was increased 10-fold. However, the effect of increased severing saturates at high severing acceleration, so filament depolymerization is always needed to some extent. Thus, we see that the observed transition kinetics can be reproduced by simultaneously introducing a moderate increase in depolymerization velocity and a more pronounced enhancement of severing activity. This is consistent with a mechanism that both fragments actin into shorter filaments and enhances pointed end depolymerization.

## Acknowledgements

We thank H Oberleithner and A Schwab for access to the AFM microscope and for help with calcium measurements. We thank Paul Hardy for comments on the manuscript. This work was supported by the German Research Foundation (SFB1009 to RWS, TV, JR and HP, KE685/4–2 to MMK and QU116/5–2 to BQ), the Cells-in-Motion Cluster of Excellence (EXC1003–CiM, FF-2015–08, University of Münster), the University of Münster (IZKF SEED 06/15 to CPD) and the Max-Planck Society. The authors declare no competing financial interests.

## Additional information

### Funding

| Funder | Grant reference number | Author |
|---|---|---|
| Deutsche Forschungsgemeinschaft | SFB1009-B10 | Hermann Pavenstädt<br>Roland Wedlich-Söldner |
| Max-Planck-Gesellschaft | | Roland Wedlich-Söldner |
| Deutsche Forschungsgemeinschaft | EXC1003-CiM | Roland Wedlich-Söldner |
| Deutsche Forschungsgemeinschaft | KE685/4-2 | Michael M Kessels |
| Deutsche Forschungsgemeinschaft | QU116/5-2 | Britta Qualmann |
| Deutsche Forschungsgemeinschaft | FF-2015-08 | Roland Wedlich-Söldner |
| University of Münster | IZKF SEED 06/15 | Christopher P Dlugos |

The funders had no role in study design, data collection and interpretation, or the decision to submit the work for publication.

### Author contributions

PW, CES, RA, JF, IG-A, BM, RW-S, Conception and design, Acquisition of data, Analysis and interpretation of data, Drafting or revising the article; AJ, CK, MW, LHK, WH, Acquisition of data, Analysis and interpretation of data; CPD, Conception and design, Acquisition of data, Analysis and interpretation of data; MS-H, wrote Matlab code and analyzed data, Analysis and interpretation of data; JK, Wrote Matlab code and analyzed data, Analysis and interpretation of data; EM, Conducted experiments and analyzed data, Acquisition of data; KCM, DEM, Acquisition of data, Analysis and

interpretation of data, Contributed unpublished essential data or reagents; AR, Acquisition of data, Contributed unpublished essential data or reagents; HNH, HP, Provided reagents and interpreted data, Contributed unpublished essential data or reagents; TV, BQ, Analysis and interpretation of data, Contributed unpublished essential data or reagents; JR, Contributed unpublished essential data or reagents; MMK, Conception and design, Analysis and interpretation of data, Drafting or revising the article, Contributed unpublished essential data or reagents

### Author ORCIDs

Ireth García-Aguilar, http://orcid.org/0000-0003-2654-042X
Roland Wedlich-Söldner, http://orcid.org/0000-0002-1364-7589

## Additional files

### Supplementary files

• Source code 1. Matlab script for automated nuclear rim detection.

• Source code 2. Matlab script for manual nuclear rim detection.

• Source code 3. Matlab script for analysis of intensity traces at nuclear rims.

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
