## [Decision Letter]

Thank you for submitting your article "Calcium-mediated actin reset (CaAR) mediates acute cell adaptations" for consideration by *eLife*. Your article has been favorably evaluated by Vivek Malhotra (Senior Editor) and three reviewers, one of whom, Mohan Balasubramanian, is a member of our Board of Reviewing Editors. Giorgio Scita (Reviewer #2), also involved in review of your submission, has agreed to reveal their identity

The reviewers have discussed the reviews with one another and the Reviewing Editor has drafted this decision to help you prepare a revised submission.

Summary:

The referees concurred that your work has uncovered an interesting phenomenon of how actin cytoskeleton undergoes a rapid and transient morphological change upon stress and how calcium mediated activation of IFN2 participates in the actin cytoskeletal reorganization (CaAR).

They all concur that you need to make a further advance on calcium regulation of IFN2 and its dual role in actin polymerization and depolymerization. Please note the major points by referee 2 (Giorgio Scita) that suggest a number of strategies and major point of referee 1 on potential ways to discern between formin competition. In the discussion between the referees, it was agreed that this is essential for publication in *eLife* and the referees believe that it can be accomplished in 2 months.

There are also concerns by all referees about the organization and presentation of the figures and have suggested ways to improve the figures.

The referees’ comments are appended verbatim for your benefit.

*Reviewer #1:*

The authors report that calcium influx in several types of mammalian cells leads to a rapid disassembly of the cortical actin, and polymerization of actin filaments at the endoplasmic reticulum (ER) via the formin INF2. The process is reversed after a short period of time, which is termed as Calcium-mediated actin reset (CaAR). This fast and reversible actin reorganization process has recently been characterized by Shao et al., 2015 in PNAS. In the current study, Wales et al. provide detailed quantification with high temporal resolution for this process, and also generate a computational model of actin homeostasis as the mechanistic basis for CaAR. Moreover, the authors expanded the understanding of this intriguing process by documenting the molecular and cellular responses after CaAR activation. It is interesting to know that CaAR precedes and plays an important role in wound healing, when the cell membrane is damaged by laser ablation. At a longer timescale, CaAR induces expression of certain set of genes, presumably important for cellular adaptation to the environmental stresses. This study is interesting and provides novel insights into our understanding of cell stress and actin dynamics.

The actin homeostasis model to explain CaAR is only supported computationally. I do not see any experimental evidence for two pools of formin (cortex vs ER) competing for G-actin. Is there a simple experiment that can be done? i.e. increasing ER associated actin polymerization even in the absence of ionophores by knock down of cortical formin?

*Reviewer #2:*

In the present work, a global and general Calcium-dependent actin dynamics (CaAR) response is described and proposed to be required to reset the extent of actin polymerization/depolymerisation needed for various actin dependent processes including PM membrane repair, cell migration and MRTF transcription. The work extends, deepens and generalizes a set of initial, recently published, similar finding by the Shivashankar's group. Like in the previous work, INF2 is shown to be the key actin nucleator mediating the CaAR response. This notwithstanding, the manuscript advances our knowledge of a so far un-described phenomenon by showing some of the important biological consequences linked to this response.

One important issue, for which little information is provided is related to the mechanisms through which INF2 is controlled by Ca++ and how it may act in promoting a concomitant depolymerization and polymerization of actin. Numerical modelling suggests that the key to these effects rests in the competition between F and G actin amounts. Yet some additional, biochemical experiment appear necessary to support this contention and verify the model:

Mechanism of action of INF2.

INF2 actin polymerization activity is shown to be key for ER-actin polymerization and the ensuing cytosolic and cortical actin de-polymerisation. It is unclear however whether the acute elevation of any localized formin would lead to similar response. Is CaAR sensitive to the formin inhibitor Smith2? In other words, would any strongly, localized (to internal organelles) and activated actin nucleator lead to global and rapid actin dynamics changes?

It is also unclear whether INF2 localization to ER is a critical requirement for the CaAR response.

It is argued that: "(Actin dynamics-CaAR responses) are consistent with a mechanism for CaAR based on simple competition between two polymerization activities for a common pool of actin monomers." However, it is also proposed that actin severing activity of INF2 might also be important for the process, but no experimental work is done to address this possibility. Numerical simulation supports the contention that the key factor of CaAR response is the result of a simple competition between two polymerization activities for a common pool of actin monomers. If this were the case than one obvious experiment to addressing this issue and verifying the validity of the numerical simulation would be the use of reconstitution of INF2 siRNA with a siRNA-resistant mutant (an experiment that has been done in the recent publication by Shao et al. PNAS 2015 thus showing its feasibility) that is impaired in severing, while it maintains its actin polymerization activity. In this respect, it was shown by Gurel et la CB, 2014 that a C-terminal region of INF2 increases severing potency by 40-fold, and that the WH2-resembling DAD motif is responsible for this increase.

How does Ca^2+^ increase leads to activation of INF2? This issue remains unaddressed. Admittedly, various possibilities might be at play and discussion of how this might happen would at least be necessary.

*Reviewer #3:*

In this study, Wales et al. report a very generic response of mammalian cells to a sudden calcium influx, like that which occurs upon injury or following the addition of ATP or a calcium ionophore. This process, which they call "Calcium-mediated actin reset" is robust, relatively reproducible across cell types, rapid (seconds) and rapidly reversed (minutes). Importantly, the team shows that the process leads to an increase in actin filaments at ER and the nuclear envelope and a near-coincident decrease in cortical actin. Strikingly, both processes can be prevented by the silencing of INF2, an ER associated Formin in HeLa cells, suggesting that INF2 is responsible for the nucleation of the ER pool. Finally, they show that there is a longer term impact of this signalling on cells, leading to an INF2 dependent change in gene expression, which appears to be largely mediated via SRF as a result of the transient change in G/F actin levels. This may help to mediate longer term changes in F-actin organisation. Overall this is a very interesting finding that will interest a wide audience.

My major criticisms of the paper are:

1) That the images used could be more compelling and more of the image data could be better presented following quantification.

2) That there is too much data and some of it could be better served by a more detailed follow up analysis. This mostly applies to Figure 6 and Figure 7, which whilst interesting do not throw much light on the process of caAR itself. Moreover, these latter experiments are not very well controlled and do not distinguish between INF2-dependent effects / the impact of actin on cortical-ER vs the actin cortex itself, and other possible effects of these perturbations.

3) Not enough has been done to test whether INF2 is really required *both* for nucleation and accelerated actin turnover as suggested. Nucleation and competition for G-actin seem simpler explanations. Whilst the model may suggest a need for accelerated turnover, the authors don't have a great tool by which to test this since the treatments (e.g. INF2 RNAi) may affect one or both processes. But, it is hard to understand how ER-localized INF2 directly induces cortical actin turnover, while enabling the accumulation of actin at the ER. Finally, given that there is a LOSS of G actin, the increased actin turnover idea doesn’t seem especially compelling.

Therefore, I would suggest:

1) Main Figures be simplified to make them less crowded, with better labels and easier to read. For example, 2E-G are very hard to understand and are not very convincing as presented. Some could be replaced with quantitative data.

2) Some supplemental data can be removed (this doesn't mean you have to remove the authors who generated it).

3) I would remove Figure 6. I would also consider removing the model and the AFM data, not of which add much to the story.

4) The authors either prove the INF2 plays a role in turnover or stick to ER-nucleation. I would suggest focusing on nucleation and keep the story simple.

With a few tweaks the paper would make a great addition to *eLife* and the literature.

---

## [Author Response]

[…]

*Summary:*

*The referees concurred that your work has uncovered an interesting phenomenon of how actin cytoskeleton undergoes a rapid and transient morphological change upon stress and how calcium mediated activation of IFN2 participates in the actin cytoskeletal reorganization (CaAR).*

*They all concur that you need to make a further advance on calcium regulation of IFN2 and its dual role in actin polymerization and depolymerization. Please note the major points by referee 2 (Giorgio Scita) that suggest a number of strategies and major point of referee 1 on potential ways to discern between formin competition. In the discussion between the referees, it was agreed that this is essential for publication in eLife and the referees believe that it can be accomplished in 2 months.*

We thank the editor and the reviewers for their very constructive comments. To address the major criticism we have now performed the following new experiments:

1) We have teamed up with the group of Britta Qualmann and Michael Kessels, who have found that INF2 interacts with calmodulin in a calcium-dependent manner. These results are now included in Figure 3 (Results subsection “CaAR is driven by INF2-mediated actin polymerization”, second paragraph and Discussion, second paragraph.

2) We have performed the rescue experiment that was suggested by reviewer 2 but have used a CRSPR/Cas9 knock out strategy rather than siRNA-mediated knock down to obtain cleaner results. Using this approach we found that reducing INF2 depolymerization activity led to a significant slowdown and reduced amplitude of actin turnover during CaAR at both, the ER and cell cortex. Our new results provide strong validation for our theoretical model.

3) Again using INF2 KO cells and rescue experiments with different INF2 isoforms, we found that both, ER-resident and cytosolic INF2 were able to support CaAR.

*There are also concerns by all referees about the organization and presentation of the figures and have suggested ways to improve the figures.*

We have addressed all of the issues raised with figure presentation.

*The referees’ comments are appended verbatim for your benefit.*

*Reviewer #1:*

[…]

*The actin homeostasis model to explain CaAR is only supported computationally. I do not see any experimental evidence for two pools of formin (cortex vs ER) competing for G-actin. Is there a simple experiment that can be done? i.e. increasing ER associated actin polymerization even in the absence of ionophores by knock down of cortical formin?*

We apologize that we didn´t make this point clear enough in our manuscript. We postulate competition between the calcium- and INF2-mediated actin polymerization and all other (mostly cortical) nucleators in the cell that are active independent of intracellular calcium, including Arp2/3 and various formins. The presence of such cortical nucleators has been extensively documented in the past. In addition, we find that cortical actin returns at the time where INF2 activity is very low (no Ca^2+^ left). Using CRSPR/Cas9 we have removed INF2 from HeLa cells and now show that cortical actin is not severely reduced by this (Figure 3). We now also provide evidence that ER localization of INF2 is not required for CaAR (rescue of INF2 KO with cytosolic isoform, Figure 3). Finally, in support of our competition model we have now included new results using rescue of the INF2 KO in HeLa cells with a depolymerization defective INF2 mutant (Figure 4). A mutant in the WH2 domain (less severing activity) leads to slower overall reaction of CaAR while maintain regulation through Ca^2+^. These results now provide quantitative validations of our theoretical assumptions (see answer to main point of reviewer 2).

*Reviewer #2:*

[…]

*One important issue, for which little information is provided is related to the mechanisms through which INF2 is controlled by Ca++ and how it may act in promoting a concomitant depolymerization and polymerization of actin. Numerical modelling suggests that the key to these effects rests in the competition between F and G actin amounts. Yet some additional, biochemical experiment appear necessary to support this contention and verify the model:*

Mechanism of action of INF2.

We thank the reviewer for this valuable comment. To address this important issue we have generated extensive new data on the mechanism of INF2-mediated actin polymerization during CaAR. We now show that INF2 binds to calmodulin in a Ca^2+^-dependent manner (Figure 3), providing a possible molecular link to its strong Ca^2+^ regulation. In addition, we have generated INF2-KO cells using CRSPR/Cas9 and performed rescue experiments with various INF2 mutants, to differentiate between polymerization and depolymerization activities of INF2. Those results are now included as Figure 4.

*INF2 actin polymerization activity is shown to be key for ER-actin polymerization and the ensuing cytosolic and cortical actin de-polymerisation. It is unclear however whether the acute elevation of any localized formin would lead to similar response. Is CaAR sensitive to the formin inhibitor Smith2? In other words, would any strongly, localized (to internal organelles) and activated actin nucleator lead to global and rapid actin dynamics changes?*

This is an interesting point. We had shown in the original manuscript that constitutive active INF2 (A1549D mutant) leads to permanent polymerization of actin at the ER indicating that Ca^2+^ signals are not required for this (Figure 3). We now added experiments to show that ER localization is not required for CaAR. In fact, the cytosolic isoform of INF2 can equally support CaAR (Figure 3). Whether INF2 has unique properties that permit its function as reversible competitor is an interesting aspect. Our results using a depolymerization-deficient INF2 variant indicate that rapid reorganization of actin might require not only Ca^2+^ activated polymerization but also Ca-activated depolymerization. Using the established KO-rescue-system we plan to further unravel the details of INF2 activities during CaAR in the future. We have also performed experiments with SMIFH2 and found that CaAR was not affected by the drug, despite the published effects of SMIFH2 on INF2. However, as we can currently not exclude cell type, time or concentration dependent effects we decided not to include these data in our manuscript.

*It is also unclear whether INF2 localization to ER is a critical requirement for the CaAR response.*

We now show that the cytosolic INF2 isoform can also support CaAR (Figure 3).

*It is argued that: "(Actin dynamics-CaAR responses) are consistent with a mechanism for CaAR based on simple competition between two polymerization activities for a common pool of actin monomers." However, it is also proposed that actin severing activity of INF2 might also be important for the process, but no experimental work is done to address this possibility. Numerical simulation supports the contention that the key factor of CaAR response is the result of a simple competition between two polymerization activities for a common pool of actin monomers. If this were the case than one obvious experiment to addressing this issue and verifying the validity of the numerical simulation would be the use of reconstitution of INF2 siRNA with a siRNA-resistant mutant (an experiment that has been done in the recent publication by Shao et al. PNAS 2015 thus showing its feasibility) that is impaired in severing, while it maintains its actin polymerization activity. In this respect, it was shown by Gurel et la CB, 2014 that a C-terminal region of INF2 increases severing potency by 40-fold, and that the WH2-resembling DAD motif is responsible for this increase.*

In the original manuscript a main conclusion of our simulations was that a simple competition model could not produce the strong and fast turnover at the cortex and ER. However, we indeed did not provide experimental validation for the predicted Ca^2+^-mediated increase in depolymerization. We now performed the experiment suggested by the reviewer. We were able to rescue INF2 knock down by siRNA with resistant INF2 constructs (not shown). However, the variable reduction of INF2 in knock down experiments makes quantitative interpretations inherently difficult. We therefore decided to completely knock out INF2 via CRSPR/Cas9 and rescue the knock out cells with the depolymerization-deficient INF2 mutant (3L) from Gurel et al. that was suggested by reviewer 2. Consistent with an important role for INF2 depolymerization during CaAR, we found a significant slowdown and reduced amplitude of turnover at both, the ER and cell cortex in the 3L mutant. The new results are provided in Figure 4. We thank the reviewer for his very helpful suggestion. Our new results provided strong validation for the theoretical model and show the strength of an interdisciplinary approach.

*How does Ca2+ increase leads to activation of INF2? This issue remains unaddressed. Admittedly, various possibilities might be at play and discussion of how this might happen would at least be necessary.*

This is an important aspect of CaAR. We now included new results showing that INF2 interacts with calmodulin in a Ca^2+^-dependent manner (Figure 3), providing a potential molecular mechanism for INF2 regulation. Whether in the cell this interaction occurs directly or via additional adaptor proteins will be a matter of future studies. We now discuss this aspect in the second paragraph of the subsection “CaAR is driven by INF2-mediated actin polymerization”.

*Reviewer #3:*

[…]

*My major criticisms of the paper are:*

*1) That the images used could be more compelling and more of the image data could be better presented following quantification.*

We have gone through all figures. We have attempted to clarify panels and improve labeling wherever possible. We have tried to include quantifications wherever possible.

*2) That there is too much data and some of it could be better served by a more detailed follow up analysis. This mostly applies to Figure 6 and Figure 7, which whilst interesting do not throw much light on the process of caAR itself. Moreover, these latter experiments are not very well controlled and do not distinguish between INF2-dependent effects / the impact of actin on cortical-ER vs the actin cortex itself, and other possible effects of these perturbations.*

We agree that we have not formally proven that the observed effects on cell protrusions are indeed dependent on INF2 or CaAR. We did however show a very tight temporal correlation and showed that all protrusions formed within seconds of the return of actin to the cell cortex. As the strong effects on cell protrusions and cell migration will likely be interesting to a wide audience and provide interesting links to the long-term effects on transcription, we decided to retain these parts in the manuscript. As discussed in our response to reviewer 2 we clearly state the weaker link between CaAR and cellular protrusions (subsection “CaAR mediates acute cellular reorganization”, first paragraph) and reduced the emphasis on this aspect (removed Figure 6—figure supplement 1).

*3) Not enough has been done to test whether INF2 is really required both for nucleation and accelerated actin turnover as suggested. Nucleation and competition for G-actin seem simpler explanations. Whilst the model may suggest a need for accelerated turnover, the authors don't have a great tool by which to test this since the treatments (e.g. INF2 RNAi) may affect one or both processes. But, it is hard to understand how ER-localized INF2 directly induces cortical actin turnover, while enabling the accumulation of actin at the ER. Finally, given that there is a LOSS of G actin, the increased actin turnover idea doesn’t seem especially compelling.*

To address this important issue that was also raised by reviewer 2 we have performed rescue experiments expressing various INF2 mutants in INF2 KO cells. These experiments have shown that cytosolic INF2 can also support CaAR and that the depolymerization activity of INF2 (residing in its WH2 domains) is essential to achieve the rapid and strong reorganization typical for CaAR (Figure 4). To keep our conclusion from the model simple, we now removed the aspects of synchrony between ER and cortex changes and on the involved G-actin levels, as suggested by the reviewer.

*Therefore, I would suggest:*

*1) Main Figures be simplified to make them less crowded, with better labels and easier to read. For example, 2E-G are very hard to understand and are not very convincing as presented. Some could be replaced with quantitative data.*

We have reorganized Figure 2 to better represent the important findings on cell freezing.

*2) Some supplemental data can be removed (this doesn't mean you have to remove the authors who generated it).*

We have removed Figure 1—figure supplement 1, Figure 3—figure supplement 1, Figure 4, Figure 4—figure supplement 1, and Figure 6—figure supplement 1.

*3) I would remove Figure 6. I would also consider removing the model and the AFM data, not of which add much to the story.*

We think these parts cover very important aspects of CaAR. With the new data on the INF2 depolymerization mutant the model has been experimentally validated and is now a much stronger component of the story (Figure 4). The AFM data nicely complements our results on cell freezing, which further underscores the reset aspect of CaAR and provides important implications in the context of PM damage and cell stress response. The panels in Figure 6 regarding a role of CaAR during PM sealing are central to the physiological role of CaAR and should definitely stay. As discussed above and in our response to reviewer 2, we also would like to keep the correlative data on cell protrusions/cell migration in the manuscript.

*4) The authors either prove the INF2 plays a role in turnover or stick to ER-nucleation. I would suggest focusing on nucleation and keep the story simple.*

We have now complemented INF2 KO HeLa cells with a depolymerization deficient INF2 mutant that strongly affects CaAR kinetics (Figure 4, see above). These new results strongly support the proposed dual role of INF2.